# SpSCO: A Speculative Sampling Approach to Neural Combinatorial Optimization

## Abstract

An open challenge in neural combinatorial optimization (CO), such as using reinforcement learning (RL) and diffusion models (DMs), is the speed–quality trade-off: sequential RL decoders generalize well but tend to settle for suboptimal tours, while DMs generate high-quality full solutions at the cost of long training and slow iterative sampling. We present **SpSCO**, a new framework inspired by speculative sampling (SpS) for large language models (LLMs) inference. Resembling SpS in LLMs, a light-weight draft model (analogous to the sequential RL decoder in SpSCO) collaborates with a high-capacity target model (analogous to a DM in SpSCO) to achieve fast, robust, and high-quality inference – the target model is triggered only when there is a "cognitive divergence" between the draft and target models or internal uncertainty of the draft model. This SpS strategy allows SpSCO to achieve high solution quality while reducing the computational overhead from DMs. Notably, SpSCO is model-agnostic and can be plug-and-play across various RL and DM backbones. It also shows strong robustness: even with under-trained, suboptimal RL and diffusion backbones, SpSCO achieves a better trade-off between performance and inference time compared with strong RL and DM backbones on diverse CO instances across various scales while attaining faster inference time on large-scale instances.

## 1 Introduction

Combinatorial optimization (CO) problems like the Traveling Salesman Problem (TSP) are a cornerstone of computational and operations research, which aim to find an optimal solution from exponentially growing combinations of candidates (Garey & Johnson, 1979; Papadimitriou & Steiglitz, 1982). While traditional exact solvers like Concorde (Applegate et al., 2006) provide optimal solutions, the exponential complexity of CO problems makes them impractical for large-scale instances. This has spurred the design of heuristic methods and, more recently, learning-based approaches.

Neural solvers have emerged as a promising paradigm, learning heuristics directly from data. The state-of-the-art (SOTA) methods broadly fall into two categories, each with a distinct trade-off. Reinforcement learning (RL) methods (Bello et al., 2016; Kool et al., 2019a) construct solutions sequentially in an autoregressive fashion. They are fast and exhibit strong generalization, but their greedy, step-by-step nature often results in suboptimal solutions, as early mistakes cannot be corrected. To mitigate this limitation, some works augment these RL frameworks with search procedures like Monte Carlo Tree Search (MCTS), but this comes at the cost of substantially increased inference cost (Fu et al., 2021b; Wang et al., 2021). Diffusion models (DMs) (Sohl-Dickstein et al., 2015; Graikos et al., 2022) emerge as the new SOTA neural CO solvers, which learn the global structure of the solution space and generate the entire solution from random noises during inference. While they produce exceptionally high-quality solutions, the iterative denoising process results in a steep computational overhead during inference.

This inherent speed-quality dilemma motivates our work. Our underlying research question is: can we synergistically combine the best of both worlds—the speed of RL models and the quality of DMs—and create a hybrid solver that is both fast and highly accurate?

We introduce **SpSCO** (pronounced as "SPESS-koh", see Figure 1), a *fast*, *robust*, *high-quality*, and *model-agnostic* inference framework for neural CO problems that is inspired by the idea of speculative sampling (SpS) (Chen et al., 2023a; Leviathan et al., 2023; Li et al., 2024; Xu et al., 2024) for

large language models (LLMs). SpSCO adaptively couples a sequential RL decoder (analogous to a light-weight draft model in LLMs SpS) with a conditional DM (analogous to a high-capacity target model in LLMs SpS). The core of our approach is an *a priori* trigger mechanism that intelligently decides *when* to invoke the powerful but expensive DM. At each step of the RL decoding process, this trigger monitors for two states, which we quantify using two complementary signals: (1) the RL policy's high internal uncertainty, measured by its entropy, and (2) the "cognitive divergence" between the RL policy and a global prior derived from the DM, measured by KL divergence.

Only when a critical juncture is detected—high RL uncertainty or significant cognitive divergence—is the DM activated. Upon activation, it performs two tasks: it corrects the immediate next step for the RL model and generates a set of complete, high-quality candidate solutions based on the RL partial one. A final selection stage then chooses the best tour from the RL-DM hybrid solution and all DM proposals. This strategic allocation of computational resources allows SpSCO to harness the DM's strengths precisely when they are most needed, avoiding its overhead in decisions.

Our experiments demonstrate the efficacy and robustness of SpSCO. Even when equipped with under-trained, suboptimal RL and diffusion backbones, SpSCO achieves a better optimality gap of 0.02% with an inference time of approximately 1 second per instance on the standard TSP-100 benchmark. On the larger TSP-500 (TSP-1000) problem, it attains a 3.56% (4.58%) gap equipped with under-trained RL and DM components, outperforming most learning-based baselines in quality while being substantially faster than other diffusion models. Besides, the performance of SpSCO on the orienteering problem (OP) is on average 3.54% higher than that of the high-quality DM backbone with the same number of rollouts across different datasets. Similarly, it is on average 1.68% higher on the maximum independent set (MIS) problem. These results underscore our central thesis: a principled, divergence-driven coordination of heterogeneous models can achieve top-tier performance and efficiency during test time, offering a more computationally effective path forward than simply scaling up models or relying on exhaustive search. Our code is included in the supplementary material for reproducibility and will be released upon paper acceptance.

## 2 RELATED WORK

**RL Solvers.** RL solvers frame CO problems as a sequential decision-making process. The seminal work of (Bello et al., 2016) applied a Pointer Network (Vinyals et al., 2015) with policy gradient methods to construct solutions step-by-step. This was advanced by Kool et al. (2019a) with the Transformer architecture that became a foundational blueprint for many subsequent models like POMO (Kwon et al., 2020), which introduced techniques to leverage multiple optima, and Sym-NCO (Kim et al., 2022), which exploited solution symmetries. NAR4TSP (Xiao et al., 2024) proposed a non-autoregressive RL algorithm, but with lower solution quality than the autoregressive versions. Later studies used RL to learn a generalized policy for certain COPs in graphs (Bengio et al., 2020; Chen et al., 2021) or proposed more advanced learning paradigms to enhance solution quality, such as searching in a continuous latent space (Chalumeau et al., 2023), optimizing policies based on preferences between solutions (Pan et al., 2025), and framing problems within game-theoretic contexts (Li et al., 2025). Recently, to further push the performance boundary, researchers have explored heavy decoder architectures (Luo et al., 2023) or integrated neural networks with iterative meta-heuristics like genetic algorithms (Kim et al., 2025a) and ant colony sampling (Kim et al., 2025b). Some RL algorithms like Invit (Fang et al., 2024) and UDC (Zheng et al., 2024) are designed for large-scale COPs, while BQ-NCO (Drakulic et al., 2023) leverages bisimulation quotienting for zero-shot generalization on massive graphs. While these models are computationally efficient and serve as strong baselines, their myopic, autoregressive construction or heavy iterative search poses a limitation, often trapping them in local optima. SpSCO uses an RL model as its fast, "*draft model*", but crucially seeks to mitigate the limitation with a global-aware component.

**Heatmap and Diffusion Models for CO.** To overcome the sequential limitations of RL solvers, another line of research has focused on non-autoregressive methods that generate a solution in a single shot, often by producing a "heatmap" of edge or node probabilities (Li et al., 2018; Fu et al., 2021a). Joshi et al. (2019) used Graph Convolutional Networks (GCNs) to predict edge inclusion probabilities for TSP. More recently, diffusion models, with their remarkable success in generative tasks, have been adapted for CO. Graikos et al. (2022) mapped TSP instances to images to be solved by a standard image diffusion model. A more direct approach, DIFUSCO (Sun & Yang, 2023),

introduced a graph-based diffusion framework that operates on the problem's native graph structure, casting it as a discrete vector generation task. This paradigm has been explored for TSP (Athaide et al., 2023) and extended to other routing problems like VRP (Chen et al., 2023b), while related score-based generative models have been proposed for a broader class of CO problems (Weinberg & Welling, 2021). These models excel at capturing the global structure of optimal solutions, leading to state-of-the-art quality. However, their reliance on a slow, multi-step iterative sampling process makes them computationally intensive. Our work incorporates a diffusion model as a costly, "*target model*", but mitigates its high inference cost through a sparse and adaptive invocation strategy.

## 3 PRELIMINARY

SpSCO is built upon two distinct neural solver paradigms: a sequential RL solver and a generative Diffusion Model. We briefly outline the standard inference process for each. The training procedure and details on the two solvers are added in the appendix A.

### 3.1 INFERENCE WITH RL SOLVERS

An RL solver decomposes the node (or edge, etc. in the CO problem) selection into a sequence and sequentially chooses one based on the trained policy network $\pi_\theta$. Given the current state in the state space, i.e. $s_k \in \mathcal{S}$, the policy $\pi_\theta$ outputs a probability distribution $\pi_\theta(\cdot|s_k)$ over the action space $\mathcal{A}$. During the inference process, the policy network is queried to determine the next action at each step $k$ until the solution is completed, e.g., when a Hamilton loop is found in a TSP. In a standard greedy decoding setting, the action $a$ with the highest probability in $\pi_\theta(\cdot|s_k)$ is chosen:

$$a_k = \arg \max_{a \in \mathcal{A}_k} \pi_\theta(a|s_k). \tag{1}$$

This greedy decoding generation of solutions is computationally fast, as it requires only $N$ forward passes of the policy network and $\arg \max$ computation, in which $N$ is the length of a solution. To improve the quality or diversity of solutions, different decoding types like sampling and beam search are proposed (Kool et al., 2019b), which introduce more computation burden and increase the solving time. Therefore, SpSCO adopts the greedy action selection in its RL decoding strategy.

### 3.2 INFERENCE WITH CONDITIONAL DIFFUSION MODELS

In contrast, a diffusion model is a non-autoregressive, generative model that learns to produce a complete solution holistically. For discrete problems like TSP, where a solution can be represented by a binary adjacency matrix $x \in \{0, 1\}^{N \times N}$, the model uses a discrete diffusion process. It consists of a denoising network, $D_\phi$, trained to reverse a noising process that gradually corrupts the solution by flipping its binary entries. Notably, this process can be conditioned on prior information, such as a given prefix of a tour in a TSP, to steer the generation.

Inference (sampling) starts with a matrix of pure random noise (e.g., a random binary matrix), $x_T$, and iteratively refines it over $T$ total steps to produce a clean solution, $x_0$. At each denoising step $t$ (different from the time step $k$ in the episode for RL), the denoising network $D_\phi$ predicts the original clean solution $\hat{x}_0$ based on the current noisy matrix $x_t$ and a conditioning prefix $c$. This prediction is then used to sample a slightly less noisy matrix $x_{t-1}$. The reverse step is generally formulated as:

$$x_{t-1} \sim p_\phi(x_{t-1}|x_t, c), t \in \{1, ..., T\}. \tag{2}$$

where $p_\phi$ denotes the learned reverse transition distribution. This iterative process allows the model to capture complex global dependencies, leading to high-quality results. However, it is computationally expensive, requiring many forward passes through the large denoising network $D_\phi$. To accelerate this, we adopt fast sampling strategies like DDIM (Song et al., 2021) to reduce the number of denoising steps while maintaining solution quality.

## 4 OUR APPROACH: SPSCO

In this section, we detail SpSCO, a framework that adapts the principles of **speculative sampling** (Chen et al., 2023a; Leviathan et al., 2023) to combinatorial optimization. In Section 4.1, we first

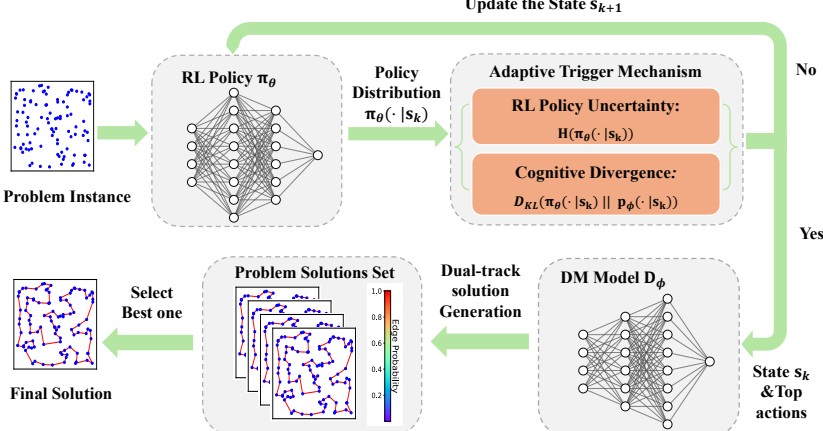

Figure 1: The inference pipeline of SpSCO. At each step, the RL policy (draft model) outputs a probability distribution over the available actions, $\pi_\theta(\cdot|s_k)$. This distribution is then evaluated against two trigger conditions: (1) RL policy's entropy, $H(\pi_\theta(\cdot|s_k))$, exceeds a threshold $H_{\text{thresh}}$, signaling high internal uncertainty; or (2) Cognitive divergence, $D_{KL}$, between the RL policy and a global prior derived from the DM, denoted as $p_\phi(\cdot|s_k)$, is above its threshold $D_{KL,\text{thresh}}$. Once triggered, DM is invoked. SpSCO begins to operate the dual-track solution generation to collect candidate solutions generated by RL and DM, and returns the best one as the final solution.

introduce the motivation from sps and the core "draft-then-verify" concept of SpSCO. The inference loop and designs of the adaptive trigger mechanism are described in Section 4.2 and 4.3. In Section 4.4, we describe the dual-track solution generation strategy initiated by this trigger, explaining how it produces both a corrected hybrid solution and a set of entirely new proposals for final selection.

## 4.1 MOTIVATION FROM SPS

Foundational work in large language models (LLMs) demonstrated that the high latency of powerful autoregressive models can be amortized by using a smaller, faster "draft model" to generate speculative candidates, which are verified in a single parallel pass by the large "target model" (Leviathan et al., 2023). SpSCO is the first framework to successfully translate this "draft-then-verify" paradigm to the structured, sequential decision-making domain of combinatorial optimization. We instantiate this by assigning distinct roles to two complementary solver families. An RL policy serves as the fast draft model, which excels at rapidly constructing a solution by proposing the next action at each step. A DM acts as the powerful target model, possessing a holistic understanding of the global solution space and effectively verifying or correcting the RL model's myopic, step-by-step decisions.

However, the true challenge is not merely to avoid the powerful target model, but to leverage its global, simulative perspective in a computationally feasible manner. To this end, SpSCO repositions the speculative framework as a **high-efficiency simulation engine**. The RL draft model proposes plausible future actions, and the DM target model evaluates them in parallel, providing the deep lookahead needed for high-quality decision-making at a fraction of the cost of traditional methods like Monte Carlo Tree Search. The overall architecture of this process is depicted in Figure 1.

The structural analogy between our approach and the standard speculative decoding is illustrated in Figure 2. As shown in the left panel, LLMs typically employ a parallel verification mechanism where a target model accepts or rejects tokens proposed by a draft model. SpSCO adapts this "draft-then-verify" paradigm to the sequential decision-making of CO, but with a strategic modification: instead of continuous verification, we employ a conditional intervention.

## 4.2 THE SPSCO INFERENCE LOOP

The SpSCO inference process is an iterative solution construction loop. At each step $k$, the RL policy generates a probability distribution over available actions ($\pi_\theta(\cdot|s_k)$). Before committing to the greedy choice, the adaptive trigger mechanism (introduced in Section 4.3) is evaluated.

**LLMs' Speculative Decoding Mechanism**

**SpSCO's Decoding Mechanism**

Figure 2: Comparison between the LLMs Speculative Decoding and the SpSCO trajectory correction framework. **Upper:** Standard speculative decoding in LLMs employs a "draft & verify" loop where a target model verifies tokens generated by a draft model. **Lower:** SpSCO adapts this paradigm for CO via a dual-track strategy. Upon detecting a critical juncture (Trigger), the Diffusion Model intervenes to generate both a corrected hybrid path and fully diffusion-generated proposals.

If the trigger conditions are not met, the RL model's locally optimal greedy action is accepted, and the partial solution is extended. However, if the trigger fires, indicating a critical juncture, SpSCO initiates its dual-track path generation strategy (in Section 4.4). This process is governed by a **single-trigger design**; once the DM is invoked at step $k^*$, a flag is set, and the RL policy completes the remainder of the hybrid path without further checks. This design ensures a predictable computational budget and maximizes the impact of the single, high-leverage intervention.

Unlike the multiple correction of speculative decoding in LLM, SpSCO adopts the single-trigger design based on both the practical implications and theoretical analysis. From the theoretical aspect, our single-trigger strategy is grounded in the theoretical optimality of minimizing total trajectory error. By treating the sequence of steps executed by the RL agent as an effective episode length $t$, established theoretical analyses in Model-Free RL (Jin et al., 2020; Ghosh et al., 2022; Velegkas et al., 2022) suggest that cumulative regret scales polynomially with the horizon. This results in a super-linear growth in cumulative error (e.g., $\propto t^{3/2}$) due to compounding variances and distributional shifts, causing the marginal risk of the RL policy to increase monotonically over time.

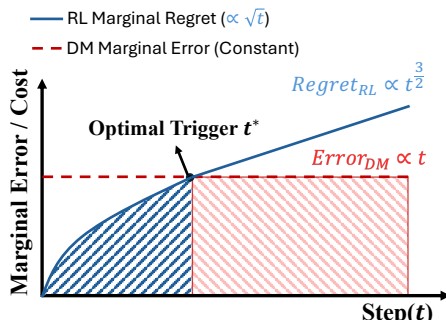

Figure 3: Schematic illustration of marginal regret dynamics. The RL agent's marginal error increases with the horizon ($\propto \sqrt{t}$), while the Diffusion Model (DM) maintains a constant error baseline.

In contrast, while the DM's cumulative regret scales linearly with the problem dimension $d$, its marginal regret remains constant relative to the step index $t$ (Benton et al., 2023) (Details in Appendix C). As shown in Fig. 3, the total accumulated error corresponds to the area under the marginal error curves (depicted by the shaded regions). Geometrically, the intersection point $t^*$ represents the precise moment where the RL's rising marginal cost exceeds the DM's constant baseline. Therefore, the strict monotonicity of the RL marginal error ensures a unique intersection point with the constant DM baseline, theoretically justifying the sufficiency of a single-trigger strategy.

Besides, from the practical aspect, activating the trigger multiple times could potentially yield high-quality solutions (similar to exhaustive search). However, doing so would cause the inference time to grow linearly with the number of triggers, destroying the efficiency advantage. The supportive experiments and analysis are in the Appendix B.5.

### 4.3 THE ADAPTIVE TRIGGER MECHANISM

The core of SpSCO is the adaptive trigger that determines *when* to speculatively invoke the DM. It is designed to activate only at critical junctures where the RL policy's decision is likely to be unreliable. We identify two such states: RL policy is internally uncertain, and its confident decision externally disagrees with the DM's global knowledge. As illustrated in Figure 1, the trigger is based on two complementary signals, and the DM is invoked if either signal surpasses a predefined threshold:

$$H(\pi_\theta(\cdot|s_k)) > H_{\text{thresh}} \quad \vee \quad D_{KL}(\pi_\theta \| p_\phi) > D_{KL,\text{thresh}} \tag{3}$$

The following subsections state the mathematical computation of each signal.

#### 4.3.1 SIGNAL 1: RL POLICY UNCERTAINTY

The first signal measures the RL model's internal uncertainty. High uncertainty suggests the RL policy does not have a single, confident choice, making it a prime moment for guidance. We quantify this uncertainty using the standard policy entropy:

$$H(\pi_\theta(\cdot|s_k)) = -\sum_{a \in \mathcal{A}_k} \pi_\theta(a|s_k) \log \pi_\theta(a|s_k) \tag{4}$$

We use policy entropy because it directly quantifies the "flatness" of the action probability distribution: a high entropy value signifies a more uniform distribution where probabilities are spread across many actions—precisely reflecting the model's indecisiveness as no single action is a clear winner.

#### 4.3.2 SIGNAL 2: COGNITIVE DIVERGENCE BETWEEN RL AND DM

The second signal measures the external disagreement, or "cognitive divergence," between the RL policy's choice and the DM's global perspective. This is designed to catch the most dangerous failure mode of a greedy policy: confidently making a locally optimal move that leads to a globally suboptimal solution. Quantifying this requires two steps: efficiently probing the DM to form a prior distribution, and then measuring the divergence between this prior and the RL's own distribution.

**Diffusion-Derived Prior via Energy Probing.** To obtain the DM's opinion without the cost of a full sampling run, we introduce a lightweight single-step denoising **energy probe**. For a set of Top-M candidate actions $\{a^{(i)}\}_{i=1}^M$ proposed by the RL at the state $s_k$, we calculate the "energy" $E_\phi(c_k^i)$ of each corresponding prefix $c_k^i = s_k \oplus a^{(i)}$. First, we construct a target adjacency matrix $A(c_k^i)$ where $A_{uv}(c_k^i) = 1$ if the edge $(u, v)$ exists in the prefix $c_k^i$, and 0 otherwise. We then perturb this target with noise to a specific timestep $t_{\text{probe}}$ to obtain $X_{t_{\text{probe}}}$. We define the "energy" $E_\phi(c_k^i)$ as the negative log-likelihood of the proposed edges being recovered by the model:

$$E_\phi(c_k^i) = -\frac{1}{|\mathcal{E}(c_k^i)|} \sum_{(u,v) \in \mathcal{E}(c_k^i)} \log \left( \sigma \left( D_\phi(X_{t_{\text{probe}}}, c_k^i)_{uv} \right) \right), \tag{5}$$

where $\mathcal{E}(c_k^i)$ is the set of edges in the partial tour $c^{(i)}$, $\sigma(\cdot)$ is the sigmoid function, and $D_\phi(\cdot)_{uv}$ is the model's predicted logit for edge $(u, v)$. Crucially, this summation acts as a mask: it specifically measures the diffusion model's confidence that the *proposed structural connections* are valid parts of a global solution, ignoring non-connected pairs.

We then convert these energy values into a DM-derived prior distribution, $q_{\text{DM}}(\cdot|s_k)$, using a Boltzmann distribution with temperature $\tau$:

$$q_{\text{DM}}(a^{(i)}|s_k) = \frac{\exp(-E_\phi(c_k^i)/\tau)}{\sum_{j=1}^M \exp(-E_\phi(c_k^j)/\tau)}. \tag{6}$$

**Quantifying Divergence.** With the RL policy $\pi_\theta(\cdot|s_k)$ and the diffusion-derived prior $q_{\text{DM}}(\cdot|s_k)$ defined over the same candidate set, we quantify their disagreement using the Kullback-Leibler (KL) Divergence:

$$D_{KL}(\pi_\theta \| q_{\text{DM}}) = \sum_{i=1}^{M} \pi_\theta(a^{(i)}|s_k) \log \frac{\pi_\theta(a^{(i)}|s_k)}{q_{\text{DM}}(a^{(i)}|s_k)}. \tag{7}$$

A high divergence signals that the RL model's confident local choice ($a \sim \pi_\theta$) contradicts the global structural patterns learned by the diffusion model ($a \sim q_{\text{DM}}$).

### 4.4 Dual-Track Solution Generation and Final Selection

Our SpSCO framework operates on a "dual-track solution generation" strategy when the adaptive trigger fires at a critical step $k^*$. The strategy initiates two parallel path generations: the hybrid path correction and full DM proposal generation.

For the hybrid path correction track, DM is used to *correct the immediate next action* for the hybrid path $\pi_H$. Instead of the RL model's greedy choice $a_{k^*}$, we select the corresponding action $a_{k^*}^{DM}$ at the step $k^*$ from the best solution in the DM's candidate pool $\mathcal{P}_{DM}$. This corrected action, $a_{k^*}^{DM}$, is appended to the tour, and the fast RL policy then autoregressively completes the remaining steps.

Simultaneously, in the full DM proposal generation track, DM is used to generate a set of complete, high-quality candidate solutions, $\mathcal{P}_{DM}$. This is done by taking the high-probability candidate actions from the RL policy at step $k^*$ and using each one to form a prefix. The conditional DM then generates a full tour for each prefix via its iterative denoising process. To govern the scope of this generation, we implement a *decoupled candidate selection strategy*. While the antecedent probing phase utilizes a fixed Top-M set to strictly maintain metric stability for the trigger, this generation phase adopts Nucleus Sampling (Top-P) to dynamically construct the candidate pool. The Top-P mechanism adaptively scales the computational budget—contracting the candidate pool during high-confidence states to conserve resources, while expanding it under uncertainty to maximize the probability of recovering the optimal solution.

After both tracks are completed, SpSCO compares the total cost of all tours in the proposal set and returns the one with the minimum cost. Algorithm 1 formalizes this entire procedure, summarizing the interplay between the adaptive trigger and the dual-track strategy. It details the step-by-step logic where the trigger is evaluated, and depending on the outcome, either a standard RL step is taken or the dual-track generation is initiated, culminating in a final selection of the best overall solution.

## 5 Experiments

We evaluate SpSCO on the 2D Euclidean Traveling Salesman Problem (TSP), a standard benchmark for neural combinatorial optimization. Our evaluation is designed to assess SpSCO's solution quality, inference speed, and the effectiveness of its core components across various problem scales.

### 5.1 Experimental Settings

**Datasets.** We use standard public datasets for TSP with 50, 100, and 500, 1000 nodes. Problem instances are generated by sampling node coordinates uniformly from the unit square $[0, 1]^2$. We use the Concorde solver (Applegate et al., 2006) to obtain optimal solutions for training our diffusion model and for calculating optimality gaps during evaluation.

**Baselines.** We compare SpSCO with a comprehensive set of methods: exact solver Concorde, heuristic solver LKH-3 (Helsgaun, 2017), and state-of-the-art learning-based approaches, including RL models (AM (Kool et al., 2019a), POMO (Kwon et al., 2020)), supervised models (GCN (Joshi et al., 2019)), and diffusion-based solvers (DIFUSCO (Sun & Yang, 2023), T2T (Li et al., 2023)).

**Implementation Details.** Our SpSCO framework orchestrates an RL policy and a conditional DM. For RL backbones, we use official pre-trained checkpoints. Our conditional DM, termed **Prefix-Difusco**, adapts the GNN architecture from DIFUSCO and is specifically trained with a masked loss and a curriculum learning strategy to complete tours from given prefixes. To convert the DM's probabilistic heatmap output into a valid tour, we employ a deterministic greedy decoding strategy. Full architectural details, training procedures, and decoding algorithms are provided in Appendix A.

---

**Algorithm 1:** SpSCO: Speculative Sampling for Combinatorial Optimization

---

**Input:** TSP instance $I$, RL policy $\pi_\theta$, DM denoiser $D_\phi$, thresholds $H_{thresh}, D_{KL,thresh}$,
Top-$P$ threshold $P_{thresh}$, Probe size $M$

**Initialize:** Hybrid path $\pi_H \leftarrow ()$, DM proposals $\mathcal{P}_{DM} \leftarrow \emptyset$, state $s_0$, $dm\_triggered \leftarrow$ false

1 **for** $k = 0, \ldots, N-1$ **do**
2      **if** *not dm_triggered* **then**              ▷ *Adaptive Trigger Evaluation*
3          Calculate RL policy uncertainty $H(\pi_\theta(\cdot|s_k))$ via Eqn. (4)
4          Select fixed Top-M actions $\{a^{(i)}\}_{i=1}^M$ from $\pi_\theta$ Calculate denoising energy
            $E_\phi(s_k \oplus a^{(i)})$ for each candidate via Eqn. (5)
5          Derive DM prior distribution $q_{DM}(\cdot|s_k)$ via Eqn (6)
6          Calculate KL divergence $D_{KL,k} \leftarrow D_{KL}(\pi_\theta||q_{DM})$ via Eqn (7)
7          **if** $H_k > H_{thresh}$ *or* $D_{KL,k} > D_{KL,thresh}$ **then**     ▷ *Dual-Track Generation*
8              Select action set $\{a^*\}$ via cumulative probability $P_{thresh}$
9              **for** *each selected action $a^*$* **do**
10                  Form prefix $c^* = s_k \oplus a^*$
11                  Generate a full tour proposal $\pi_{DM} \sim$ DM_Sampler$(D_\phi, c^*)$ and add to $\mathcal{P}_{DM}$
12              **end**
13              Set $dm\_triggered \leftarrow$ true
14              Select $a_k^{DM}$ from the best solution in $\mathcal{P}_{DM}$ as the RL action: $a_k \leftarrow a_k^{DM}$
15          **else**
16              $a_k \leftarrow \arg\max_a \pi_\theta(a|s_k)$        ▷ *Not trigger DM: RL greedily select*
17          **end**
18      **else**
19          $a_k \leftarrow \arg\max_a \pi_\theta(a|s_k)$             ▷ *After DM triggered: RL finish $\pi_H$*
20      **end**
21      Append $a_k$ to hybrid path $\pi_H$ and update state to $s_{k+1}$
22 **end**
23 Calculate cost $C(\pi_H)$ and all costs in $\mathcal{P}_{DM}$             ▷ *Final Selection*
24 **return** $\arg\min_{\pi \in \{\pi_H\} \cup \mathcal{P}_{DM}} C(\pi)$

---

**Evaluation Metrics.** We report three key metrics: the average tour length, the percentage optimality gap to the exact solver's solutions, and the average inference time per problem instance.

### 5.2 MAIN RESULTS

To validate the performance and plug-and-play nature of our model-agnostic framework, we integrated SpSCO with two different RL backbones, the Attention Model (AM) and POMO. Furthermore, we demonstrate the generalizability of SpSCO to other combinatorial tasks in Appendix B.4, specifically the OP (Appendix B.1) and the MIS (Appendix B.2).

#### 5.2.1 PERFORMANCE ON TSP-50 AND TSP-100

Table 1 presents the performance of SpSCO and baseline methods on TSP-50 and TSP-100 instances. The results are categorized by whether a **2-opt** local search post-processing step is applied. The 2-opt algorithm (Croes, 1958) is a classic and effective local search heuristic for the TSP, which iteratively improves a tour by removing two edges and reconnecting the two resulting paths in the only other possible way to see if the new tour is shorter. It is important to first clarify that the Prefix_Difusco model listed in the table is our custom version of Difusco, which we specifically adapted to handle partial tours (prefixes) as conditions, making it compatible with the SpSCO framework.

#### 5.2.2 PERFORMANCE ON LARGE-SCALE TSP

SpSCO's superior speed-quality trade-off becomes even more pronounced on large-scale problems, as shown in Table 2. On TSP-500, using a DM-undertrained that was only briefly fine-tuned for 20 epochs, SpSCO achieves a **3.56%** optimality gap in just **1.20 mins**. This result is not only significantly better in quality than other RL-based methods and standalone DM solvers like T2T (5.09%),

Table 1: Results with **Greedy Decoding** on TSP-50 and TSP-100. RL: Reinforcement Learning, SL: Supervised Learning, G: Greedy Decoding, S: Sample Decoding. * denotes results that are quoted from previous works. The "Prefix_Difusco (16 sample)" represents the best result obtained by running our Prefix_Difusco model sampling 16 times in parallel.

| ALGORITHM | TYPE | TSP-50 | | TSP-100 | |
|---|---|---|---|---|---|
| | | LENGTH ↓ | DROP ↓ | LENGTH ↓ | DROP ↓ |
| Concorde (Applegate et al., 2006) | Exact | 5.69 | 0.00% | 7.76 | 0.00% |
| 2Opt (Croes, 1958) | Heuristics | 5.86 | 2.95% | 8.03 | 3.54% |
| AM* Kool et al. (2019a) | RL+G | 5.80 | 1.76% | 8.12 | 4.53% |
| GCN* Joshi et al. (2019) | SL+G | 5.87 | 3.10% | 8.41 | 8.38% |
| Transformer* Bresson & Laurent (2021) | RL+G | 5.71 | 0.31% | 7.88 | 1.42% |
| POMO* Kwon et al. (2020) | RL+G | 5.73 | 0.64% | 7.84 | 1.07% |
| Sym-NCO* Kim et al. (2022) | RL+G | 5.73 | 0.64% | 7.84 | 0.94% |
| Image Diffusion* Graikos et al. (2022) | SL+G | 5.76 | 1.23% | 7.92 | 2.11% |
| DIFUSCO ($T_s$=50) Sun & Yang (2023) | SL+G | 5.71 | 0.45% | 7.85 | 1.21% |
| T2T ($T_s$=50,$T_t$=15) Li et al. (2023) | SL+G | 5.69 | 0.07% | 7.77 | 0.20% |
| **Prefix_Difusco** (Used in SpSCO) | SL+G | 5.70 | 0.27% | 7.81 | 0.62% |
| **Prefix_Difusco** (16 sample) | SL+S | 5.69 | 0.04% | 7.76 | 0.04% |
| **SpSCO (AM + DF)** | **RL+SL+G** | 5.69 | **0.03%** | 7.76 | **0.02%** |
| **SpSCO (POMO + DF)** | **RL+SL+G** | 5.69 | **0.03%** | 7.76 | **0.02%** |
| AM | RL+G+2OPT | 5.77 | 1.41% | 8.02 | 3.32% |
| GCN | SL+G+2OPT | 5.77 | 1.40% | 8.01 | 3.21% |
| Transformer | RL+G+2OPT | 5.70 | 1.06% | 7.96 | 1.89% |
| POMO | SL+G+2OPT | 5.73 | 0.63% | 7.91 | 1.62% |
| Sym-NCO | SL+G+2OPT | 5.73 | 0.64% | 7.90 | 0.76% |
| DIFUSCO | SL+G+2OPT | 5.69 | 0.09% | 7.78 | 0.22% |
| T2T | SL+G+2OPT | 5.69 | **0.02%** | 7.76 | 0.06% |
| **SpSCO** | **RL+SL+G+2OPT** | 5.69 | 0.03% | 7.76 | **0.01%** |

but also faster than high-performance competitors like DIFUSCO (5.70m) and T2T (4.90m). This good performance scales to even larger instances. On TSP-1000, even when pairing an RL model directly from the TSP-500 checkpoint with another similarly undertrained DM, SpSCO delivers a state-of-the-art optimality gap of **4.58%** among learning-based methods. It achieves this result in just **2.43 mins**, over 6 times faster than T2T. This efficiency stems directly from our core design: the fast RL model handles the majority of decisions, while the computationally intensive DM is invoked only for a strategic, high-impact correction. This allows SpSCO to scale effectively, delivering top-tier solutions without the prohibitive runtime of exhaustive search or full iterative generation.

## 5.3 ABLATION STUDY

We conduct ablation studies on TSP-100 to validate SpSCO's core components and design choices.

**Importance of the Trigger Mechanism.** As shown in Table 3, our dual-signal trigger is essential. Relying solely on either the entropy or the KL divergence trigger leads to significantly worse optimality gaps (0.05% and 0.09%, respectively). This confirms that both internal policy uncertainty (entropy) and external disagreement with the DM (KL divergence) are complementary and necessary signals for making effective intervention decisions.

**Trigger Behavior and Sensitivity.** Our analysis reveals a clear trade-off landscape for the trigger thresholds. The entropy threshold directly balances solution quality and inference time (Table 14 in Appendix B.4), while a stricter (lower) KL divergence threshold proves superior for performance (Table 15 in Appendix B.4). Notably, with our default thresholds, the trigger fires in 100% of instances at a very early average step of 0.52. This supports our rationale of "strategic early intervention": SpSCO identifies the initial steps as the most critical, correcting the RL model's trajectory before errors can propagate. The full analysis, including sensitivity to candidate exploration strategies and the robustness of the energy probe, is detailed in Appendix B.4.

Table 2: Results on large-scale TSP problems. RL, SL, AS, G, and S denote Reinforcement Learning, Supervised Learning, Active Search, Greedy decoding, and Sample decoding, respectively. Len means the average tour length. * indicates the baseline for computing the performance gap. The TIME column reports the average inference time per instance.

| ALGORITHM | TYPE | TSP-500 | | | TSP-1000 | | |
|---|---|---|---|---|---|---|---|
| | | LENGTH↓ | DROP↓ | TIME | LENGTH↓ | DROP↓ | TIME |
| Concorde | Exact | 16.55* | 0.00% | 37.66m | 23.53* | 0.00% | 6.65h |
| Gurobi | Exact | 16.55 | 0.00% | 45.63h | 23.53 | 0.00% | 48h |
| LKH-3 (default) | Heuristics | 16.55 | 0.00% | 46.28m | 23.53 | 0.00% | 2.57h |
| AM | RL+G | 20.02 | 20.99% | **0.47s** | 28.52 | 21.21% | 1.09s |
| GCN | SL+G | 29.72 | 79.61% | 6.67m | 43.15 | 83.38% | 28.52m |
| POMO(greedy) | RL+G | 19.13 | 15.62% | 0.49s | 27.15 | 15.38% | **0.94s** |
| POMO(Massive Sampling) | RL+S | 18.59 | 12.32% | 2.97m | 28.34 | 20.44% | 3.25m |
| DIMES | RL+G | 18.93 | 14.38% | 0.97m | 27.23 | 15.73% | 2.08m |
| DIMES | RL+AS+G | 17.81 | 7.61% | 2.10h | 25.11 | 6.72% | 4.49h |
| LEHD | RL/SL+G | 16.82 | 1.64% | 0.59s | 24.29 | 3.23% | 3.84s |
| DIFUSCO | SL+G | 18.11 | 9.41% | 5.70m | 25.68 | 9.14% | 11.5m |
| T2T | SL+G | 17.39 | 5.09% | 4.90m | 25.17 | 8.87% | 15.66m |
| Prefix_Difusco(used in spsco) | SL+G | 17.92 | 8.23% | 0.50m | 25.86 | 9.91% | 0.38m |
| Prefix_Difusco(16 samples) | SL+S | 17.31 | 4.58% | 2.10m | 24.96 | 6.07% | 5.94m |
| **SpSCO(POMO)** | **RL+SL+G** | 17.24 | 3.56% | 1.20m | 24.61 | 4.58% | 2.43m |
| **SpSCO(LEHD)** | **RL+SL+G** | **16.81** | **1.57%** | 0.73m | **24.24** | **3.04%** | 0.36m |
| POMO(greedy) | RL+G+2OPT | 17.77 | 7.37% | 0.58s | 25.10 | 6.67% | 1.05s |
| DIMES | RL+G+2OPT | 17.65 | 6.62% | 1.01m | 24.83 | 7.38% | 2.29m |
| DIMES | RL+AS+G+2OPT | 17.31 | 4.57% | 2.10h | 24.33 | 5.22% | 4.49h |
| LEHD | RL/SL+G+2OPT | 16.77 | 1.40% | 1.83s | 24.06 | 2.29% | 3.85s |
| DIFUSCO | SL+G+2OPT | 16.83 | 1.68% | 5.75m | 23.92 | 1.66% | 17.52m |
| T2T | SL+G+2OPT | 16.86 | 1.92% | 2.42m | 23.86 | 1.42% | 15.90m |
| **SpSCO(POMO)** | **RL+SL+G+2OPT** | 16.82 | 1.63% | 2.32m | 23.92 | 1.66% | 4.40m |
| **SpSCO(LEHD)** | **RL+SL+G+2OPT** | **16.74** | **1.2%** | 0.98m | **23.64** | **0.46%** | 0.36m |

Table 3: Ablation study on the core components of SpSCO, Trigger Mechanism, evaluated on the TSP-100 dataset. Gap (%) is the optimality gap compared to the Concorde solver. Time (s) is the average inference time per instance.

| Model / Method | Description | Gap (%)↓ |
|---|---|---|
| Entropy-Only | Policy entropy trigger | 0.05% |
| KL-Only | KL divergence trigger | 0.09% |
| **Full Model** | **Full Model** | **0.02%** |

# 6 CONCLUSION

We introduced SpSCO, a novel speculative sampling framework that synergistically combines a fast, sequential RL model with a high-quality, nonautoregressive diffusion model for solving combinatorial optimization problems. The core of our contribution is a lightweight trigger mechanism that uses policy entropy and KL divergence to adaptively invoke the diffusion model at critical decision points. This "cognitive divergence" metric, calculated via an efficient single-step energy probe, effectively identifies when the local, greedy decisions of the RL model begin to deviate from the global optimum manifold learned by the diffusion model. Our results on standard TSP benchmarks demonstrate that SpSCO achieves state-of-the-art solution quality while being more computationally efficient than other high-performance methods. This work opens up a promising new direction in neural CO solvers: better neural CO solvers may lie not just in developing larger or more complex models, but in the principled and intelligent hybridization of diverse, complementary approaches.

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

## A  IMPLEMENTATION DETAILS OF PREFIX_DIFUSCO AND RL MODELS

Here we provide detailed specifications for our backbone conditional diffusion model, Prefix_Difusco, used in the experiments. About the RL model, we uniformly use the code and environment provided by the RL4CO Berto et al. (2023) package for training [1]. The detailed hyperparameters are listed in A.4.

### A.1  MODEL ARCHITECTURE

The core of Prefix_Difusco modifies the GNN architecture from DIFUSCOSun & Yang (2023) for the conditional generation task. The key components are:

- **Node Feature Embedding**: Node coordinates are first converted into high-dimensional features using a sinusoidal positional embedding. A binary feature indicating whether a node is part of the prefix is concatenated to this embedding. A final linear layer projects this combined feature vector to the GNN's expected input dimension ($d_{node} = 128$).
- **PrefixEncoder**: An LSTM-based encoder takes the sequence of node features corresponding to the prefix tour and outputs a single global conditioning vector ($d_{cond} = 256$). This vector summarizes the properties of the given partial tour.
- **DifuscoGNNEncoder**: This is the main denoising network. It is a 12-layer GNN that processes a graph where nodes have the features described above. At each layer, the GNN's message passing is conditioned by both the global prefix vector from the `Prefix_Encoder` and a sinusoidal embedding of the current timestep $t$.
- **Output Head**: The GNN outputs a logit for each potential edge in the graph, representing the probability of that edge being part of the optimal tour.

### A.2  TRAINING PROCEDURE

The model is trained to predict the ground-truth adjacency matrix $x_0$ from a noised version $x_t$ and a conditional prefix.

- **Dataset and Conditioning**: The training dataset consists of TSP instances and their optimal tours. For each sample, we derive a training instance by randomly selecting a prefix of length $k$ from the optimal tour.
- **Diffusion Process**: We use a discrete diffusion process over $T = 1000$ steps with a cosine noise schedule to corrupt the ground-truth adjacency matrix $x_0$ into a noisy matrix $x_t$.
- **Masked Loss Function**: The model's objective is to minimize the Binary Cross-Entropy (BCE) loss between the predicted adjacency matrix and the ground truth $x_0$. Crucially, the loss is only computed on the "suffix" edges—that is, all edges except those whose both endpoints are within the given prefix. This forces the model to learn how to best complete the tour.
- **Curriculum Learning**: To improve convergence and performance, we employ a multi-stage curriculum. The training starts with a distribution of long prefixes (e.g., $k \in [60, 90]$), making the completion task easier. As training progresses, the distribution of $k$ shifts towards shorter, more difficult prefixes (e.g., $k \in [1, 30]$), allowing the model to gradually master the full conditional generation task.

### A.2.1  TRAINING FOR TSP-50

The `Prefix_Difusco` model for TSP-50 was trained from scratch. We employed a 5-stage curriculum learning strategy designed to gradually increase the task difficulty. Each stage was trained for 10 epochs, initializing from the best checkpoint of the previous stage.

- **Stage 1 (Easy):** Trained on long prefixes with lengths $k \in [30, 49]$.
- **Stage 2 (Medium):** Trained on prefix lengths $k \in [10, 30]$.

---

[1] https://github.com/ai4co/rl4co

- **Stage 3 (Hard):** Trained on the full range of prefix lengths $k \in [1, 49]$.

- **Stage 4 (Short Focus):** Focused on short prefixes with lengths $k \in [1, 20]$.

- **Stage 5 (Very Short Focus):** Further focused on very short prefixes with $k \in [1, 10]$ to enhance performance on early-step decisions.

### A.2.2 TRAINING FOR TSP-100

The model for TSP-100 was also trained from scratch following a similar multi-stage curriculum learning approach. The prefix length ranges for each stage were adjusted proportionally for the larger problem size to ensure a smooth learning progression from easy to hard completion tasks. The core hyperparameters, such as hidden dimensions and learning rate, were kept consistent with the TSP-50 model.

- **Stage 1 (Easy):** Trained on long prefixes with lengths $k \in [61, 99]$.

- **Stage 2 (Medium):** Trained on prefix lengths $k \in [30, 60]$.

- **Stage 3 (Hard):** Trained on the full range of prefix lengths $k \in [1, 99]$.

- **Stage 4 (Short Focus):** Focused on short prefixes with lengths $k \in [1, 20]$.

- **Stage 5 (Very Short Focus):** Further focused on very short prefixes with $k \in [1, 10]$ to enhance performance on early-step decisions.

### A.2.3 TRAINING FOR TSP-500

Due to the significantly larger scale of TSP-500, we adopted a more advanced training strategy combining **transfer learning** and a tailored curriculum.

- **Transfer Learning**: The TSP-500 model was not trained from scratch. Instead, it was initialized using the weights from our best-trained TSP-100 checkpoint. Weights were transferred for all layers with matching names and shapes (e.g., GNN layers, prefix encoder), providing a strong starting point and accelerating convergence.

- **Curriculum Learning**: After initialization, the model was fine-tuned on TSP-500 data using a 5-stage curriculum similar to the one for TSP-50, but with ranges adjusted for $N = 500$ (e.g., Stage 1: $k \in [50, 100]$, Stage 2: $k \in [20, 50]$, etc.). For the results reported in this paper, we ran an accelerated training schedule of approximately 20 epochs in total, focusing on the most critical curriculum stages to balance performance and computational cost.

### A.2.4 TRAINING FOR TSP-1000

For the largest scale, TSP-1000, we continued the strategy of combining transfer learning with a specialized curriculum to manage the increased complexity and computational demands.

- **Transfer Learning**: The TSP-1000 model was initialized with the weights from our best-performing TSP-100 checkpoint. This transfer learning approach provided a robust feature foundation, significantly accelerating the training convergence on the larger graph size.

- **Curriculum Learning**: Following initialization, the model was fine-tuned on the TSP-1000 dataset using a 4-stage curriculum over a total of 25 epochs. The training began with easier tasks (completing tours from long prefixes, with k up to 500) and progressively moved to more difficult scenarios, focusing on shorter prefixes (k down to 1) in later stages to refine the model's ability to make critical early decisions.

### A.3 DECODING ALGORITHMS

To convert the probabilistic heatmap output from our `Prefix_Difusco` model into a valid TSP tour, we employ a deterministic greedy decoding strategy inspired by DIFUSCO (Sun & Yang, 2023). This approach ensures that given a heatmap, the resulting tour is always the same, which

is crucial for the stability of the SpSCO framework. The core decoding process follows a principled, multi-stage procedure designed to construct high-quality tours while respecting the prefix constraints.

The decoding algorithm proceeds as follows:

- **Edge Score Calculation:** The raw adjacency probability matrix $P$ from the diffusion model is first symmetrized to ensure consistency ($P' = (P + P^T)/2$). To favor shorter edges, which are fundamental to good TSP solutions, we compute an edge score for each potential edge $(i, j)$ by dividing its symmetrized probability by its Euclidean distance: $S_{ij} = P'_{ij}/\text{dist}(i, j)$. All possible edges are then sorted in descending order based on these scores.

- **Enforce Prefix Constraint:** Before any greedy selection, the decoder first enforces the given conditional prefix. All edges that form the given partial tour are mandatorily included in the solution set. A Union-Find data structure is initialized, and the degrees of the prefix nodes are updated accordingly to ensure these edges are fixed.

- **Greedy Spanning Path Construction:** The algorithm iterates through the globally sorted list of edges. For each candidate edge, it performs three checks:
    1. It is not an existing prefix edge.
    2. Adding the edge will not result in any node having a degree greater than two.
    3. Adding the edge will not form a premature cycle (verified using the Union-Find data structure).

  If all conditions are met, the edge is added to the solution set, and the node degrees and Union-Find structure are updated. This process continues until a total of $N - 1$ edges have been selected, forming a spanning path of all nodes.

- **Tour Finalization:** Once a spanning path of $N - 1$ edges is formed, there will be exactly two nodes with a degree of one (the endpoints of the path). The final edge connecting these two endpoints is deterministically added to close the path and form a valid Hamiltonian cycle. The final list of $N$ edges is then converted into a sequential tour starting from the first node of the original prefix (or node 0 if no prefix was given).

### A.4 HYPERPARAMETERS

This subsection provides a comprehensive summary of the hyperparameters for the core models employed in the SpSCO framework: The configurations for our conditional diffusion model (`Prefix_Difusco`), the Attention Model (AM), and POMO Kool et al. (2019a); Kwon et al. (2020) are detailed in Table 4, Table 5, and Table 6, respectively. To ensure rigorous comparability, we strictly adopted the architectural configurations (e.g., layers, dimensions) from the original DIFUSCO and RL4CO implementations, thereby isolating the performance gains to the SpSCO framework logic rather than backbone scaling. Besides, for inference-specific parameters such as DDIM sampling steps, values were selected based on a preliminary analysis of the speed-quality trade-off.

Table 4: Hyperparameters for Prefix_Difusco across different problem sizes.

| Parameter | TSP-50 | TSP-100 | TSP-500 | TSP-1000 |
|---|---|---|---|---|
| *Model Architecture* | | | | |
| Node Count (N) | 50 | 100 | 500 | 1000 |
| sparse factor(K) | N.A | N.A | N.A | 100 |
| GNN Layers (L) | 12 | 12 | 12 | 12 |
| Hidden Dimension | 256 | 256 | 256 | 256 |
| Node Embedding Dim | 128 | 128 | 128 | 128 |
| Prefix Condition Dim | 256 | 256 | 256 | 256 |
| *Diffusion Process* | | | | |
| Timesteps (T) | 1000 | 1000 | 1000 | 1000 |
| Beta Schedule | cosine | cosine | cosine | cosine |
| Inference Steps | 10 | 10 | 50 | 50 |
| Inference Sampler | DDIM | DDIM | DDIM | DDIM |
| *Training* | | | | |
| Batch Size (per GPU) | 128 | 96 | 4 | 8 |
| Epoch | 50 | 50 | 20 | 25 |
| Training data (per epoch) | 1500000 | 1500000 | 128000 | 65000 |
| Learning Rate | 2e-4 | 2e-4 | 2e-5 | 1e-4 |
| Optimizer | Adam | Adam | Adam | Adam |
| Training Method | Curriculum | Curriculum | Transfer&Curriculum | Transfer&Curriculum |
| Environment | 2x NVIDIA A40 GPUs | | | |

Table 5: Hyperparameters for Attention Model across different problem sizes. Need to notice, AM and POMO models for TSP-500 are trained on the tsp-200 instances and then generalized into the TSP 500.

| Parameter | TSP-50 | TSP-100 | TSP-500 |
|---|---|---|---|
| *Model Architecture* | | | |
| GNN Layers (L) | 3 | 3 | 3 |
| Hidden Dimension | 512 | 512 | 512 |
| Node Embedding Dim | 128 | 128 | 128 |
| Attention heads | 8 | 8 | 8 |
| Baseline | rollout | rollout | critic |
| *Training* | | | |
| Batch Size (per GPU) | 512 | 512 | 1024 |
| Training data (each epoch) | 1280000 | 1280000 | 1280000 |
| Epoch | 100 | 100 | 120 |
| Learning Rate | 1e-4 | 1e-4 | 1e-4 |
| LR Scheduler | multistep LR, gamma=0.1, milestone = [80, 95] | | |
| Normalization | batch | | |
| Environment | 1x NVIDIA A40 GPUs | | |

Table 6: Hyperparameters for POMO across different problem sizes.

| Parameter | TSP-50 | TSP-100 | TSP-500 |
|---|---|---|---|
| *Model Architecture* | | | |
| GNN Layers (L) | 6 | 6 | 6 |
| Hidden Dimension | 512 | 512 | 512 |
| Node Embedding Dim | 128 | 128 | 128 |
| Attention heads | 8 | 8 | 8 |
| Augment | 8 | 8 | 8 |
| *Training* | | | |
| Batch Size (per GPU) | 512 | 512 | 256 |
| Training data (each epoch) | 100000 | 100000 | 100000 |
| Epoch | 400 | 800 | 400 |
| Learning Rate | 1e-4 | 1e-4 | 1e-4 |
| LR Scheduler | multistep LR, gamma=0.1, milestone = [80, 95] | | |
| Normalization | instance | | |
| Environment | 1x NVIDIA A40 GPUs | | 2x NVIDIA A40 GPUs |

## B    ADDITIONAL EXPERIMENT AND ABLATION STUDY RESULTS

### B.1    GENERALIZABILITY TO THE ORIENTEERING PROBLEM

While our primary analysis focuses on TSP, the SpSCO framework is inherently problem-agnostic. To demonstrate this versatility, we extended our evaluation to the Orienteering Problem (OP) across varying scales (OP-50, OP-100, and OP-200), as summarized in Table 7. We explicitly benchmarked SpSCO against both the standalone RL backbone (AM) and the Diffusion backbone (prefix_Coexpander) under an equivalent inference budget ($s = 8$ rollouts). The results reveal a consistent trend: SpSCO effectively synergizes the strengths of its components, outperforming the performance of both baselines.

This advantage becomes increasingly pronounced as problem complexity grows. Remarkably, on the most challenging OP-200 benchmark, the performance gap is dramatic. While the standalone RL policy and the Diffusion backbone ($s = 8$) struggle with optimality gaps of 13.05% and 4.15%, respectively, SpSCO reduces this error to a near-optimal **1.081%**. This represents an order-of-magnitude improvement over the RL baseline and a significant boost over the pure Diffusion model, confirming that our hybrid mechanism—not just increased sampling—drives the performance.

Crucially, these substantial gains in solution quality are achieved with manageable computational overhead. SpSCO delivers these superior solutions within a few seconds (e.g., 2.7s for OP-200), remaining significantly faster than exact solvers like Gurobi (300s). This successful application to a different, complex routing problem validates the plug-and-play nature of our framework, demonstrating its ability to construct a robust solver from weaker components without the need for retraining.

### B.2    GENERALIZABILITY TO THE MAXIMUM INDEPENDENT SET PROBLEM

To validate the versatility of SpSCO beyond routing tasks, we extended our evaluation to the Maximum Independent Set (MIS) problem, a fundamental graph covering challenge. As detailed in Table 8, we benchmarked SpSCO against the specialized RL baseline (LWD) and the Diffusion backbone (prefix_COExpander) under an identical inference budget ($S = 5$). This setup allows us to rigorously assess whether the performance gains stem from our hybrid mechanism rather than simply increased computational resources.

The results demonstrate a strong synergy similar to our findings in routing problems. SpSCO consistently outperforms both learning-based baselines across all datasets, with the advantage being most pronounced on the challenging ER-700-800 benchmark. While the standalone RL policy suffers a significant 13.16% optimality gap and the pure Diffusion model plateaus at 9.16%, SpSCO effec-

Table 7: Performance comparison across OP-50, OP-100, and OP-200 benchmarks. Score denotes the average collected prize (higher is better). Gap indicates the percentage deviation from the optimal solution (lower is better). Time is the average inference time in seconds. S: Sample Decoding. $S_{cand}$ is the candidate number of full DM proposals $\mathcal{P}_{DM}$

| Method | OP-50 | | | OP-100 | | | OP-200 | | |
|---|---|---|---|---|---|---|---|---|---|
| | Score ↑ | Drop ↓ | Time ↓ | Score ↑ | Drop ↓ | Time ↓ | Score ↑ | Drop ↓ | Time ↓ |
| Gurobi-300 | 14.37 | 0.00% | 300.0s | 32.10 | 0.00% | 300.0s | 44.38 | 0% | 300.0s |
| Gurobi-30 | 14.31 | 0.40% | 30.0s | 31.13 | 3.02% | 30.0s | 31.37 | 29.31% | 30.0s |
| AM (RL) | 11.92 | 17.04% | **0.03s** | 28.37 | 11.62% | **0.03s** | 38.59 | 13.05% | **0.09s** |
| prefix_Coexpander (Greedy) | 12.27 | 14.61% | 0.09s | 28.64 | 10.77% | 0.15s | 41.36 | 6.80% | 0.19s |
| prefix_Coexpander(s=8) | 12.42 | 14.01% | 0.12s | 29.05 | 9.49% | 0.21s | 42.54 | 4.15% | 0.47s |
| prefix_Coexpander(s=64) | 12.51 | 12.94% | 1.08s | 29.20 | 9.03% | 0.89s | 43.15 | 2.77% | 2.67s |
| **SpSCO**($S_{cand}$=8) | **13.23** | **7.93%** | 0.55s | **29.51** | **8.06%** | 0.41s | **43.90** | **1.08%** | 2.70s |

tively combines their strengths to achieve a much lower gap of **5.76%**. Similarly, on RB-Large, SpSCO reduces the RL error by nearly two-thirds (from 15.31% to 5.28%), proving that inserting the DM at the critical juncture yields superior structural decisions compared to blind sampling.

SpSCO offers a superior trade-off between quality and speed compared to traditional solvers. While exact or heuristic solvers like KaMIS and Gurobi require substantial runtime (roughly 30 to 60 seconds) to compute solutions, SpSCO delivers competitive results within just **2 to 6 seconds**.

Table 8: Performance comparison across RB-LARGE, ER-700-800, and SATLIB benchmarks for the Maximum Independent Set (MIS) problem. Obj. denotes the average node number of independent sets (higher is better). Gap indicates the percentage deviation from the optimal solution (lower is better). Time is the average inference time in seconds.

| METHOD | RB-LARGE | | | ER-700-800 | | | SATLIB | | |
|---|---|---|---|---|---|---|---|---|---|
| | Obj.↑ | Drop↓ | Time↓ | Obj.↑ | Drop↓ | Time↓ | Obj.↑ | Drop↓ | Time↓ |
| KaMIS | 43.00 | 0.00% | 56.97s | 44.97 | 0.00% | 60.75s | 425.95 | 0.00% | 24.37s |
| Gurobi | 42.19 | 1.83% | 33.84s | 38.78 | 13.75% | 60.49s | 425.92 | 0.01% | 13.47s |
| LWD (RL) | 36.67 | 15.31% | **0.85s** | 39.05 | 13.16% | **0.64s** | 421.80 | 1.04% | 0.66s |
| prefix_Coexpander (Greedy) | 40.05 | 6.84% | 1.15s | 40.18 | 10.65% | 0.71s | 421.84 | 0.96% | **0.25s** |
| prefix_Coexpander (s=5) | 40.06 | 6.83% | 1.22s | 40.85 | 9.16% | 0.96s | 424.17 | 0.42% | 0.91s |
| prefix_Coexpander(s=64) | 40.58 | 5.63% | 4.48s | 40.91 | 9.03% | 4.03s | 424.78 | 0.27% | 4.27s |
| **SpSCO** ($S_{cand}$=5) | **40.73** | **5.28%** | 2.69s | **42.38** | **5.76%** | 4.69s | **425.40** | **0.12%** | 6.31s |

To validate the efficacy of our hybrid mechanism, we constrain both SpSCO and the baseline Diffusion model to an identical inference budget $S$. For the baseline, $S$ represents the number of independent sampling trajectories initiated from scratch (or a fixed prefix). For SpSCO, $S_{cand}$ represents the number of top-ranked RL actions selected at the trigger step to be completed by the Diffusion model. Thus, any performance gain observed in SpSCO is attributable to the quality of the guided exploration rather than an increase in computational resources.

## B.3 EVALUATION ON TSPLIB BENCHMARK

We further evaluate the zero-shot generalization capability of SpSCO on the TSPLIB benchmark, which comprises real-world instances with distributions distinct from the uniform data used for training. Specifically, we utilize the diffusion model backbone trained solely on TSP-100 instances to solve all TSPLIB instances, regardless of their size (ranging from 51 to 200 nodes). As summarized in Table 9, SpSCO achieves a competitive average optimality gap of 1.90%. Notably, our method successfully recovers the exact optimal solutions (0.00% gap) for 6 diverse instances (e.g., st70, kroC100, u159) and maintains a negligible gap ($< 0.5\%$) on 7 others. While performance fluctuates on specific outliers like berlin52 due to extreme structural shifts, the overall results demonstrate SpSCO's robust transferability to unseen problem scales and distributions without any fine-tuning.

Table 9: Tour length results on Original distances for TSP

| Instance | Concorde Length | SPSCO Length | Gap (%) |
|---|---|---|---|
| eil51 | 428.872 | 430.890 | 0.47 |
| berlin52 | 7544.366 | 8341.575 | 10.57 |
| st70 | 677.110 | 677.110 | 0.00 |
| eil76 | 544.369 | 544.369 | 0.00 |
| pr76 | 108159.438 | 108502.984 | 0.32 |
| rat99 | 1219.244 | 1273.394 | 4.44 |
| kroA100 | 21285.443 | 22116.191 | 3.90 |
| kroB100 | 22139.075 | 22886.367 | 3.38 |
| kroC100 | 20750.763 | 20750.762 | 0.00 |
| kroD100 | 21294.291 | 21622.426 | 1.54 |
| kroE100 | 22068.759 | 22172.182 | 0.47 |
| rd100 | 7910.396 | 7910.395 | 0.00 |
| eil101 | 640.212 | 640.212 | 0.00 |
| lin105 | 14382.996 | 14499.562 | 0.81 |
| pr107 | 44301.684 | 45478.461 | 2.66 |
| pr124 | 59030.736 | 59075.676 | 0.08 |
| bier127 | 118293.524 | 119938.688 | 1.39 |
| pr136 | 96770.924 | 96875.828 | 0.11 |
| pr144 | 58535.222 | 58698.148 | 0.28 |
| ch150 | 6530.903 | 6533.812 | 0.04 |
| kroA150 | 26524.863 | 27337.197 | 3.06 |
| kroB150 | 26127.358 | 26173.672 | 0.18 |
| pr152 | 73683.641 | 75642.734 | 2.66 |
| u159 | 42075.670 | 42075.668 | 0.00 |
| rat195 | 2333.873 | 2350.973 | 0.73 |
| d198 | 15808.652 | 17451.734 | 10.39 |
| kroA200 | 29369.407 | 30274.914 | 3.08 |
| kroB200 | 29440.412 | 30297.166 | 2.91 |
| | | **Average Gap:** | **1.90** |

## B.4 ABLATION STUDY

Therefore, SpSCO provides a flexible blueprint for creating next-generation hybrid neural solvers. By simply swapping the problem-specific RL and DM backbones, our framework can be readily extended to tackle a diverse set of complex optimization challenges, effectively navigating the speed-quality trade-off across various domains. This subsection provides detailed results and analysis for the ablation studies presented in the main paper, all conducted on the TSP-100 benchmark. These studies are designed to be self-contained, offering a deeper understanding of SpSCO's internal mechanisms and validating our design choices. Echoing the central thesis from our introduction, the following experiments empirically demonstrate how the principled, divergence-driven coordination of the RL and DM models allows SpSCO to effectively navigate the speed-quality trade-off. We specifically analyze the necessity of our dual-criteria trigger (the heart of our "cognitive divergence" measure), the critical balance between exploration breadth and computational efficiency, and the overall robustness of the framework's components. To maintain consistency with the main text, table numbering in this appendix begins at 6.

Table 10 examines the sensitivity of our framework to the DM energy probe's timestep (`dm_probe_timestep`). The results demonstrate remarkable robustness: across a wide range of timesteps from 100 to 900, the optimality gap remains stable at 0.02%. This is a significant practical advantage, as it indicates that the single-step energy probe is a reliable signal that does not require meticulous hyperparameter tuning.

Table 11 investigates the impact of the RL candidate pool size (`probe_rl_top_m`) used for the KL divergence calculation. This experiment highlights the importance of providing a sufficiently large set of candidate actions for the probe. With too few candidates (e.g., 5), the KL divergence is not a reliable indicator, resulting in a poor optimality gap (1.30%) and a low trigger rate (43.0%). Our

Table 10: Sensitivity analysis on the DM energy probe timestep, dm_probe_timestep.

| Energy Probe Ts | Gap (%) ↓ | Time (s) | Ave. trigger step (Trig. Rates) |
|---|---|---|---|
| 100 | 0.02 | 2255.95s | 0.66 (100.0%) |
| 300 | 0.02 | 2262.54s | 0.66 (100.0%) |
| **500** | 0.02 | **2238.88s** | 0.52 (100.0%) |
| 700 | 0.02 | 2246.30s | 0.51 (100.0%) |
| 900 | 0.02 | 2243.36s | 0.51 (100.0%) |

default setting of 15 ensures that the divergence metric is calculated over a meaningful distribution, allowing for effective detection of critical junctures.

Table 11: Sensitivity analysis on the RL candidate pool size, probe_rl_top_m, used for KL divergence calculation.

| Probe_RL_Top_M | Gap (%) ↓ | Time (s) | Ave. trigger step (Trig. Rates) |
|---|---|---|---|
| 5 | 1.30 | 2894.19s | 4.05 (43.0%) |
| 10 | 0.15 | 1635.85s | 5.20 (97.1%) |
| 15 | 0.02 | 2238.88s | 0.52 (100.0%) |
| 20 | 0.01 | 3025.94s | 0.00 (100.0%) |

Finally, Table 12 analyzes the effect of the candidate selection threshold for DM exploration (TopN_cum_Th). This parameter controls the breadth of the DM's search once it is triggered. The results show a clear trade-off: a small threshold (e.g., 0.2) is faster but often fails to find a high-quality solution, yielding a suboptimal 0.12% gap. Increasing the exploration breadth is crucial for capitalizing on the DM's generative power. Our default value of 0.8 allows the DM to explore a diverse set of high-probability candidates, which is vital for discovering the near-optimal paths that lead to our state-of-the-art 0.02% gap.

Table 12: Ablation study on the candidate selection strategy for DM exploration, n_cumulative_threshold.

| TopN_cum_Th | Gap (%) ↓ | Time (s) | Ave. trigger step (Trig. Rates) |
|---|---|---|---|
| 0.2 | 0.12 | 724.87s | 0.52 (100.0%) |
| 0.4 | 0.12 | 729.66s | 0.52 (100.0%) |
| 0.5 | 0.08 | 925.61s | 0.52 (100.0%) |
| 0.6 | 0.05 | 1205.19s | 0.52 (100.0%) |
| 0.8 | 0.02 | 2238.88s | 0.52 (100.0%) |

Table 13 provides a comprehensive analysis of our core trigger mechanism. The results clearly show that the dual-criteria trigger, which combines both policy entropy and KL divergence, is essential for top performance. Relying solely on the KL-divergence or entropy trigger leads to significantly worse optimality gaps (0.09% and 0.05%, respectively). This confirms our hypothesis that policy uncertainty (entropy) and cognitive divergence (KL) are complementary signals. The former identifies when the RL agent is indecisive, while the latter detects when it is confidently wrong. Together, they form a robust and effective condition for invoking the diffusion model.

Table 13: Complete Ablation study on the core components of SpSCO, evaluated on the TSP-100 dataset. Gap (%) is the optimality gap compared to the Concorde solver. Time (s) is the average inference time per instance. Avg. Trigger Step Index indicates the average step number in the tour construction at which the Diffusion Model was first invoked. Trigger Rate denotes the percentage of instances where the DM was triggered at least once. The best-performing model is highlighted in bold.

| Model / Method | Description: Triggers on | Gap (%) ↓ | Time (s) | Ave. trigger step (Trig. Rates) |
|---|---|---|---|---|
| *Trigger Mechanism Ablation* | | | | |
| SpSCO (Entropy) | policy entropy only | 0.05 | 2435.33s | 2.36 (99.8%) |
| SpSCO (KL) | KL divergence only | 0.09 | 1670.84s | 1.35 (97.0%) |
| **SpSCO (Full Model)** | **Full Model (Entropy + KL)** | **0.02** | 2238.88s | 0.52 (100.0%) |

Table 14: Sensitivity to the Policy Entropy Threshold

| Threshold | Gap (%) ↓ | Time (s) | Trigger step (Rates) |
|---|---|---|---|
| 1.4 | **0.01%** | 3031.26s | 0 (100%) |
| 1.6 | 0.02% | 2238.88s | 0.52 (100.0%) |
| 1.8 | 0.05% | **1367.75s** | 1.47 (99.4%) |
| 2.0 | 0.09% | 1508.01s | 1.44 (97.7%) |

Table 15: Sensitivity to the KL-Divergence Threshold

| Threshold | Gap (%) ↓ | Time (s) | Trigger step (Rates) |
|---|---|---|---|
| 8 | **0.02%** | **2238.88s** | 0.52 (100.0%) |
| 10 | 0.02% | 2284.97s | 0.83 (100.0%) |
| 12 | 0.03% | 2304.25s | 1.11 (100.0%) |
| 14 | 0.03% | 2334.44s | 1.32 (100.0%) |

## B.5   PERFORMANCE COMPARISON: SINGLE TRIGGER VS. MULTIPLE TRIGGERS

Table 16 presents a comparative analysis between the single-trigger and multiple-trigger strategies. While the multiple-trigger approach yields a marginally lower optimality gap (e.g., an improvement of only 0.01% on TSP-100 and 0.11% on TSP-1000), it incurs a prohibitive computational cost. Specifically, the inference time for the multiple-trigger strategy is approximately $10\times$ to $20\times$ slower on small-scale instances and $6\times$ slower on large-scale instances compared to the single-trigger counterpart.

This empirical evidence aligns with our theoretical finding in Appendix C, which posits that a single optimal handover point exists. The multiple-trigger strategy essentially performs a redundant search beyond this optimal point, offering diminishing returns in solution quality while sacrificing the efficiency that SpSCO aims to achieve. Therefore, the single-trigger strategy strikes the most favorable balance between solution quality and computational efficiency, making it the default configuration for our framework.

Table 16: Performance and runtime comparison between single and multiple trigger strategies on TSP benchmarks. Gap indicates the optimality gap (lower is better), and Time denotes the average inference time.

| Setting | TSP-50 | | TSP-100 | | TSP-500 | | TSP-1000 | |
|---|---|---|---|---|---|---|---|---|
| | Gap↓ | Time↓ | Gap↓ | Time↓ | Gap↓ | Time↓ | Gap↓ | Time↓ |
| Single | 0.03% | 0.25s | 0.02% | 1.2s | 3.56% | 1.20m | 4.58% | 2.43m |
| Multiple | 0.02% | 2.38s | 0.01% | 23.28s | 1.37% | 6.43m | 4.47% | 14.22m |

## C  THEORETICAL ANALYSIS: DIMENSION-DEPENDENCE OF REGRET

In this section, we analyze the theoretical properties of the proposed diffusion-based and RL solvers. By contrasting the linear regret scaling of the Diffusion Model with the polynomial scaling of the RL agent, we establish the existence of a unique, theoretically optimal trigger point for SpSCO. Specifically, we investigate how the "Regret"—defined as the performance gap between the generated solution and the optimal solution—scales with respect to the problem dimension $d$ for the DM and the horizon length $H$ for the RL agent. We begin by recalling the convergence bound of diffusion models established in recent literature.

**Lemma 1** (**Convergence of Diffusion Models, Theorem 1 in Benton et al. (2023)**). *Let $q_\delta$ be the target data distribution and $p_{t_N}$ be the sampling distribution of the diffusion model at the final timestep $t_N$. Under the assumption of $L^2$-accurate score estimation and finite second moments of the data distribution, the KL divergence between the approximate and target distributions is bounded by:*

$$KL(q_\delta || p_{t_N}) \leq \epsilon_{score}^2 + \kappa^2 dN + \kappa dT + de^{-2T} \tag{8}$$

*where $\epsilon_{score}$ is the score matching error, $N$ is the number of steps, and $T$ is the total time.*

**Remark 1.** Lemma 1 implies that the distributional distance scales linearly with the data dimension, i.e., $KL(q_\delta || p_{t_N}) = \mathcal{O}(d)$.

**Theorem 2** (**Linear Scaling of DM's Regret**). *Let $x \in \mathbb{R}^d$ be the decision variable generated by the diffusion model. We define the Regret $\mathcal{R}(p_{t_N}, q_\delta)$ as the absolute difference in the expected objective function $f(x)$ between the generated distribution $p_{t_N}$ and the target distribution $q_\delta$:*

$$\mathcal{R}_{DM}(p_{t_N}, q_\delta) = \left| \mathbb{E}_{x \sim p_{t_N}}[f(x)] - \mathbb{E}_{x \sim q_\delta}[f(x)] \right| \tag{9}$$

*Assume the following conditions hold:*

*1. Local Distributional Regularity (Gaussianity): In the high-density regions of interest, the distributions $p_{t_N}$ and $q_\delta$ are assumed to be locally unimodal and approximable by multivariate Gaussians with bounded covariance spectra. Specifically, we assume the covariance matrices $\Sigma$ have bounded eigenvalues, i.e., $\lambda_{\max}(\Sigma) \leq C_\Sigma < \infty$.*

*2. Regularity of Objective Function: The objective function $f(x)$ is twice differentiable. Its Hessian $H = \nabla^2 f(x)$ at the optimal mean $\mu_q$ has a bounded spectrum, i.e., $\lambda_{\max}(H) \leq L$ for some constant $L > 0$.*

*Then, the Regret scales linearly with the dimension $d$:*

$$\mathcal{R}_{DM}(p_{t_N}, q_\delta) \leq \mathcal{O}(d) \tag{10}$$

*Proof.* Due to the assumption of local distributional regularity, we model the decision variable using the multivariate reparameterization trick as $x(\epsilon) = \mu + \Sigma^{1/2}\epsilon$, where $\Sigma^{1/2}$ is a matrix square root (e.g., Cholesky factor) and $\epsilon \sim \mathcal{N}(0, I_d)$.

Let the target distribution be parameterized by $\phi_q = (\mu_q, \Sigma_q)$ and the approximate distribution by $\phi_p = (\mu_p, \Sigma_p)$, where $\Sigma_q$ and $\Sigma_p$ are general positive semi-definite covariance matrices.

First, we perform a second-order Taylor expansion of the expected objective $\mathbb{E}[f(x)]$ around the mean $\mu$. For any random variable $x = \mu + \delta$ with $\mathbb{E}[\delta] = 0$ and $\text{Cov}[\delta] = \Sigma$:

$$\mathbb{E}[f(x)] \approx f(\mu) + \mathbb{E}[\nabla f(\mu)^T \delta] + \frac{1}{2}\mathbb{E}[\delta^T H(\mu)\delta] \tag{11}$$

Since $\mathbb{E}[\delta] = 0$, the first-order term vanishes. Using the trace property $\mathbb{E}[\delta^T A\delta] = \text{Tr}(A\mathbb{E}[\delta\delta^T])$, the second-order term becomes $\frac{1}{2}\text{Tr}(H(\mu)\Sigma)$. Thus:

$$\mathbb{E}[f(x)] \approx f(\mu) + \frac{1}{2}\text{Tr}(H(\mu)\Sigma) \tag{12}$$

Now, we expand the Regret $\mathcal{R}_{DM}$. We approximate $f(\mu_p)$ around the optimal mean $\mu_q$. Since $\mu_q$ corresponds to the target (optimal) distribution, we assume $\nabla f(\mu_q) \approx 0$:

$$f(\mu_p) \approx f(\mu_q) + \frac{1}{2}(\mu_p - \mu_q)^T H(\mu_q)(\mu_p - \mu_q) \tag{13}$$

Substituting these expansions into the definition of Regret:

$$\mathcal{R}_{DM} = \left| \left( f(\mu_p) + \frac{1}{2}\text{Tr}(H_p\Sigma_p) \right) - \left( f(\mu_q) + \frac{1}{2}\text{Tr}(H_q\Sigma_q) \right) \right|$$

$$\approx \frac{1}{2} \left| (\mu_p - \mu_q)^T H_q(\mu_p - \mu_q) + \text{Tr}(H_q(\Sigma_p - \Sigma_q)) \right| \tag{14}$$

By Assumption 2, the spectral norm of the Hessian is bounded by $L$ (i.e., $\|H_q\|_2 \leq L$). Using the trace inequality $\text{Tr}(AB) \leq \|A\|_2 \text{Tr}(B)$ for positive semi-definite matrices, and noting that the trace is the sum of eigenvalues:

$$\mathcal{R}_{DM} \leq \frac{L}{2} \left( \|\mu_p - \mu_q\|^2 + |\text{Tr}(\Sigma_p - \Sigma_q)| \right) \tag{15}$$

Since $\text{Tr}(\Sigma) = \sum_{i=1}^{d} \lambda_i(\Sigma)$ and we assume bounded eigenvalues (Assumption 1), the trace term scales linearly with $d$ (i.e., $\text{Tr}(\Sigma) = \mathcal{O}(d)$).

We compare this to the KL divergence for general multivariate Gaussians:

$$KL(q_\delta \| p_{t_N}) = \frac{1}{2} \left( \text{Tr}(\Sigma_p^{-1}\Sigma_q) - d + (\mu_p - \mu_q)^T \Sigma_p^{-1}(\mu_p - \mu_q) + \ln\frac{|\Sigma_p|}{|\Sigma_q|} \right) \tag{16}$$

Assuming well-conditioned covariance matrices, the KL divergence is similarly dominated by trace terms and quadratic forms involving the means, which also scale with $d$. Finally, invoking Lemma 1, since $KL(q_\delta \| p_{t_N}) = \mathcal{O}(d)$, and $\mathcal{R}_{DM}$ shares the same dimensional scaling structure (dominated by trace and squared error sums), it follows that:

$$\mathcal{R}_{DM}(p_{t_N}, q_\delta) \leq \mathcal{O}(d) \tag{17}$$

$\square$

**Remark 2** (Applicability to Neural Combinatorial Optimization). The assumptions posited in Theorem 2 are inherently aligned with the mathematical formulation of combinatorial optimization (CO) problems under **continuous relaxation**. In diffusion-based solvers (e.g., DIFUSCO), the decision variable $x$ represents a continuous relaxation of discrete decisions, acting as a probabilistic "heatmap" (e.g., $x \in [0,1]^{N \times N}$ for TSP or $x \in [0,1]^N$ for MIS).

Crucially, regarding the Regularity of Objective Function (Condition 2), the regret analysis considers the continuous extension of the problem objective $f(x)$, which typically satisfies the smoothness and bounded Hessian conditions:

1. Linear Relaxation (e.g., TSP, Shortest Path): The relaxed objective is often linear with respect to the decision variable (e.g., $f(x) = \sum_{i,j} w_{ij}x_{ij}$). Since the second derivative of a linear function is zero, the Hessian is the zero matrix, which trivially satisfies the bounded spectrum condition ($L = 0$).

2. Quadratic Relaxation (e.g., MIS, MaxCut): Problems formulated as Quadratic Unconstrained Binary Optimization (QUBO) possess a natural quadratic relaxation $f(x) = x^T Q x$. The Hessian is a constant matrix $2Q$. For finite graphs, its spectral radius is finite, satisfying the bounded spectrum condition.

Consequently, Theorem 2 provides a theoretical justification that the performance regret (in terms of the actual optimization objective) scales linearly with the problem dimension $d$, mitigating the curse of dimensionality.

While the diffusion model's regret is primarily characterized by the problem dimension, the performance of the Reinforcement Learning agent is fundamentally governed by the horizon length $H$. Established theoretical analyses in Model-Free RL with linear function approximation (Jin et al., 2020; Ghosh et al., 2022; Velegkas et al., 2022) indicate that the cumulative regret scales polynomially with the horizon, rather than linearly. Specifically, due to compounding variances and distributional shifts over sequential decisions, the cumulative regret $\mathcal{R}_{RL}(H)$ is bounded by $\mathcal{O}(H^{3/2})$ or even more.

This super-linear growth implies that the *marginal* risk of the RL policy—the instantaneous error added at each subsequent step $t$—is not constant. By differentiating the cumulative regret scaling with respect to the effective horizon, we observe that the marginal error at step $t$ scales as:

$$\mathcal{R}_{RL}^{mar}(H) \propto \frac{d}{dH} H^{3/2} \propto \sqrt{H} \tag{18}$$

As illustrated in Figure 3, this results in a monotonically increasing marginal error curve for the RL agent. In contrast to the diffusion model's constant marginal error (Theorem 2), the RL agent becomes progressively less reliable as the episode length increases. The total accumulated error corresponds to the area under the marginal error curves (depicted by the shaded regions). Geometrically, the intersection point $t^*$ represents the precise moment where the RL's rising marginal cost exceeds the DM's baseline. Switching policies anywhere else would inevitably increase the total shaded area, leading to suboptimal performance. Consequently, SpSCO is designed to detect this specific theoretical minimum—where error propagation renders the RL policy less reliable than the DM—and execute a single, decisive handoff to ensure the optimal solution quality.

## D   STATEMENT ON THE USE OF LARGE LANGUAGE MODELS (LLMS)

During the preparation of this manuscript, we utilized a Large Language Model (LLM) as a general-purpose writing assistance tool. The use of the LLM was strictly limited to improving the quality and clarity of the English prose.

Specifically, the LLM was employed for the following tasks:

- Proofreading to identify and correct typographical errors.
- Correcting grammatical mistakes and ensuring syntactical correctness.
- Rephrasing sentences to improve readability, flow, and conciseness.

The core scientific ideas, theoretical derivations, experimental design, results, and conclusions presented in this paper were conceived and articulated entirely by the human authors. The LLM did not contribute to any aspect of the research ideation or the generation of the scientific content. Its role was exclusively that of a language editing and refinement tool.

