# OpenReview forum: "SpSCO: A Speculative Sampling Approach to Neural Combinatorial Optimization"
_ICLR.cc/2026/Conference — ICLR 2026 Conference Desk Rejected Submission_

### Official Review · Reviewer_sc5a · 2025-10-24

**Soundness:** 3
**Presentation:** 1
**Contribution:** 3
**Rating:** 2
**Confidence:** 4

**Summary:**

This paper proposes SpSCO, a speculative sampling framework for neural combinatorial optimization (CO). Drawing inspiration from speculative decoding in large language models (LLMs), the authors design an inference pipeline that combines a fast sequential RL “draft model” with a conditional diffusion “target model”.
At each decoding step, SpSCO uses an adaptive trigger mechanism based on policy entropy (internal uncertainty) and KL divergence (cognitive disagreement) to decide whether to invoke the diffusion model. When triggered, a dual-track generation strategy corrects the immediate step of the draft model and generates full alternative tours via the target model, from which the best solution is chosen.

The method is benchmarked on multiple TSP datasets and shows strong performance: near-optimal results on small instances (TSP-100) and improved speed–quality trade-offs on large instances (TSP-500/1000). Ablations highlight the value of the dual-trigger mechanism.

**Strengths:**

- **Novel and Well-Motivated Framework:**
    The paper presents a creative and conceptually well-grounded adaptation of speculative sampling—originally designed for large language model inference—to the domain of neural combinatorial optimization. The analogy between draft and target models in LLMs and the RL and diffusion components here is intuitive and intellectually appealing. Even though the implementation details need clarification, the underlying idea of using uncertainty-guided coordination between a fast heuristic and a strong generative solver is novel and worth exploring further.

- **Compelling Use of Cognitive Divergence:**
    The introduction of cognitive divergence (a KL-based measure between the RL and DM policies) as a lightweight coordination signal is conceptually elegant. Leveraging the diffusion model’s energy-derived prior to quantify disagreement provides a principled and computationally efficient way to trigger corrections, which could inspire broader applications in hybrid model inference.

- **Comprehensive Empirical Evaluation:**
    The paper evaluates its framework on a wide range of TSP instances (50, 100, 500, and 1000 nodes) and supplements these with results on the Orienteering Problem (OP-100), demonstrating promising scalability. The reported results achieve near-optimal performance on smaller instances (e.g., a 0.02% optimality gap on TSP-100) and remain competitive on larger problems, indicating practical potential once the methodology is clarified.

- **Model-Agnostic and Extensible Design:**
    The proposed framework is compatible with different RL backbones (Attention Model, POMO), and the design principles appear general enough to extend to other combinatorial optimization settings. This plug-and-play property—if validated—would make SpSCO a flexible inference layer for future hybrid neural solvers.

- **Extensive Ablation and Sensitivity Analyses:**
    The appendix includes detailed ablations on the entropy/KL trigger mechanism, energy probe timesteps, candidate pool sizes, and cumulative threshold settings. These analyses suggest the approach is reasonably robust to hyperparameter choices and provide valuable insight into how each component contributes to final performance.

- **Potential for Broad Impact if Clarified:**
    Despite its current inconsistencies, the paper’s central concept—combining fast RL guidance with diffusion-based refinement under a speculative coordination policy—is innovative and relevant to both the combinatorial optimization and generative modeling communities. With a clearer, internally consistent description and fairer benchmarking, this line of work could represent a meaningful step toward more efficient neural solvers.

**Weaknesses:**

- The paper contains irreconcilable contradictions in describing its core method, making it impossible to understand or implement SpSCO from the paper alone:
    1. Figure 2 vs. Single-Trigger Design: Figure 2 depicts multiple DM interventions throughout tour construction (corrections at steps 3, 5, and 7), directly contradicting the text's explicit statement of a "single-trigger design" where "once the DM is invoked at step $k^*$, a flag is set, and the RL policy completes the remainder of the hybrid path without further checks" (lines 139-141). This makes the comparison to LLM speculative sampling misleading, as speculative decoding verifies multiple tokens iteratively, not just once.

    2. Algorithm 1 Logical Errors: The outer ```if```-statement (line 4: ```if tour not complete AND not dm_triggered```) lacks a corresponding ```else``` clause. After ```dm_triggered``` is set to true (line 19), this condition becomes ```false``` for all remaining iterations, yet line 26 attempts to append ```a_next``` and update the state. Without an else clause showing how RL continues after DM invocation, the algorithm appears to terminate prematurely.

    3. Text Contradictions: Section 4.4 describes selecting "the action from the candidate pool that is most preferred by the DM's prior" (lines 312-314), suggesting the correction uses the energy-based prior $p_\phi$. However, Algorithm 1 line 20 suggests extracting the action from complete DM proposals. These are fundamentally different approaches.

    4. Conceptual Framework Mismatch: The paper frames SpSCO as involving a "propose-verify-correct cycle" at "each step" (lines 128-129), but the single-trigger design means only **one** step receives DM intervention. The entire "speculative sampling" analogy from LLMs, where draft tokens are continuously verified, does not actually apply to SpSCO's one-shot intervention strategy.

    Required corrections / improvements the authors must provide:
        - Consistency in describing the method in the main text, algorithm and figure.
        - A corrected Algorithm 1 with complete logic flow after ```dm_triggered = true```.
        - A revised Figure 2 showing only one DM intervention
        - Consistent description of how the corrected action is selected (prior-based vs. proposal-based).
        - Clarification of whether this should even be called "speculative sampling" given the single intervention.


    These are not minor presentation issues but fundamental communication failures that prevent readers from understanding, evaluating, or reproducing the method. The empirical results may be valid, but without algorithmic clarity, the contribution cannot be properly assessed.



- The single-trigger design appears ad hoc and conceptually inconsistent with the goal of speculative coordination: it assumes only one “crucial” step in the entire RL rollout requires correction. Table 11 further shows that the diffusion model is almost always triggered extremely early ($k^∗=0.5$), meaning the DM intervenes after the first or second step and effectively constructs the rest of the tour. This raises the question of what purpose the RL component serves if its output is overridden almost immediately. To demonstrate that the hybrid rollout is meaningful, the authors should report winning rates — i.e., how often the final chosen tour comes from the RL-completed hybrid path versus a DM proposal. Without such evidence, the “speculative” RL-DM collaboration appears superficial, and the framework behaves more like a slightly modified diffusion solver than a true hybrid system.

- Benchmark comparisons seem rather unfair. As far as I understand are all models except SpSCO evaluated based on a single, greedy rollout, while SpSCO performs multiple rollouts due to its "dual track generation", from which the best solution is selected. To account for that, other models should be allowed to sample multiple rollouts as well.

- Following on from that, Table 10 reveals that the optimality gap decreases monotonically as the cumulative probability threshold, and thus the number of diffusion rollouts, increases, accompanied by higher runtime. This is the standard sample-quality trade-off already observed in probabilistic NCO solvers: more samples yield better tours but slower inference. The purported benefit of SpSCO’s “speculative” design therefore appears to derive mainly from performing additional diffusion rollouts from multiple prefixes, rather than from the proposed entropy/KL trigger or the hybrid inference logic itself. For instance, the DIFUSCO paper reports a 6-percentage-point reduction in optimality gap simply by enabling sampling-based decoding. Therfore, the paper should disentangle improvements purely due to increased sampling from those due to the speculative mechanism and report compute-matched comparisons.


- Weak Support for Generality: Experiments are limited to simple routing problems (TSP and Orienteering Problem). Evaluation on more complex routing and/or scheduling problems would better undermine the models generality.

**Questions:**

- please clarify and comment on the issues raised under weaknesses.
- Can you provide ablation on the single-trigger design vs. allowing multiple triggers?
- why is the Attention Model slower than SpSCO? In my understanding, this cannot be the case, since the AM only invokes a single, greedy rollout with a similar architecture as the draft model of SpSCO, but without the dual trigger mechanism and DM generation.

---

> ### Author Response · Authors · 2025-11-24
> **Response to Weakness 1 and 2**
>
> **Weakness 1:  Clarifications (Exist Contradictions)**
>
> **Response:**
>
> We sincerely appreciate the reviewer for very carefully reading our manuscript and pointing out the inconsistencies and the clarity issues. We have meticulously revised the manuscript to ensure alignment between the text, figure, and algorithm.
>
> Summary of the updates:
>
> To eliminate ambiguity, we first clarify the intended "Single-Trigger, Dual-Track" workflow: The RL agent constructs the solution autoregressively until a specific step $t^\*$, where the trigger fires exactly once. At this point, SpSCO splits into two tracks:
>
> *  Hybrid Path: The DM provides a single-step correction ($a_{k^\*}^{DM}$), and the RL agent exclusively completes the remainder of the tour.
>
> *  DM Proposals: The DM generates full solutions in parallel based on the current prefix. The final solution is selected from these candidates.
>
> 1. Figure 2 vs. Single-Trigger Design: We have redrawn Figure 2 in the revision. It now accurately depicts the one-shot intervention logic described in the summary: RL drafts $\to$ Trigger fires once at $t^\*$ $\to$ DM intervenes $\to$ RL completes the rest. The reason why we chose a single trigger is explained in the global response.
>
> 2. Algorithm 1 Logical Errors & Presentation: We have corrected Algorithm 1 to explicitly include the else clause: if dm_triggered is true, the algorithm now correctly proceeds to the standard autoregressive step to complete the tour. Simultaneously (addressing feedback on conciseness), we have condensed the pseudo-code, abstracting the detailed mathematical calculations of Entropy/Energy into the main text to focus strictly on the corrected dual-track decision logic.
>
> 3. Text Contradictions (Selection Strategy): We have revised Section 4.4 to unify this description on Page 6, Lines 311-312. We select the corresponding action $a_{k^\*}^{DM}$ at the step $k^\*$ from the best solution in the DM's candidate pool $\{\pi_{DM}\}$, ensuring that the narrative aligns with the corrected Algorithm 1.
>
> 4. Conceptual Framework ("Speculative Sampling" Naming): While SpSCO adopts a "one-shot" intervention, we believe the term "Speculative Sampling" is justified by the shared **"Draft-then-Verify"** philosophy.
>
> * **Why we use the name**: We respectfully argue that SpSCO adheres to the fundamental definition of Speculative Sampling introduced by [1]: the decoupling of generation into a fast Draft Model (our RL agent) and a high-quality Target Model (our Diffusion Model). The core mechanism—speculating a future trajectory with a cheap model and accepting/correcting it with a strong verifier—remains the heart of our framework.
>
> * **Why "One-shot"**: The reason why we chose the single trigger is explained in the part of global response named **"Practical and theoretical rationale of the single-trigger design"**.
>
> [1] Chen C, Borgeaud S, Irving G, et al. Accelerating large language model decoding with speculative sampling[J]. arXiv preprint arXiv:2302.01318, 2023.
>
>
> **Weakness 2:  The Meaning of RL Existence Caused by the Too-Early Triggered DM.**
>
> **Response:**
>
> We respectfully disagree that the single-trigger design is "ad hoc" or that the RL collaboration is "superficial." We defend the rationale of SpSCO through theoretical optimality and empirical adaptivity:
>
> 1. Theoretical Basis for Single-Trigger (Refuting "Ad Hoc"):
>
> The theoretical analysis of why we select the single trigger is explained in the part of global response, named ‘Practical and theoretical rationale of the single-trigger design’. The concrete details are in Section 4.2 of the revised version.
>
> 2. RL Utility and Adaptivity (Refuting "Superficiality"):
>
> Regarding the concern that RL is "overridden immediately" (based on TSP results), we clarify that the trigger is highly adaptive to the problem structure rather than fixed. The early intervention in TSP ($k^* \approx 0.5$) reflects that problem's extreme sensitivity to initial nodes. To demonstrate that the RL component is structurally essential, we present our new results on the Orienteering Problem (OP), where the average trigger step for OP-50, OP-100, and OP-200 is 3.46, 1.79, and 20.06, respectively. As for the MIS problem, the average trigger steps for rblarge, er700-800, and Satlib are at step 56.14, 10.09, and 81.77, respectively. The fact that the RL agent independently navigates over 20 steps (10% of the tour) in OP-200 proves it is not superficial; it acts as an effective "fast launcher" that autonomously handles the computationally cheaper parts of the trajectory. The trigger correctly identifies that OP requires later intervention than TSP, validating SpSCO as a true hybrid system that adapts to instance difficulty.

---

> > ### Author Response · Authors · 2025-11-24
> > **Response to Weakness 3, 4 and 5**
> >
> > **Weakness 3:  Unfair Comparison of Multiple Sample Rollouts**
> >
> > **Response:**
> >
> > We agree with the reviewer that comparing a multi-rollout method against single-rollout baselines requires careful consideration of the computational budget. To ensure a fair comparison, we have explicitly included a multi-sample baseline for our strongest competitor, the Diffusion Model, in our experiments.
> >
> > As shown in Table 1 (Prefix_Difusco (16 samples)), we evaluated the diffusion baseline with 16 parallel rollouts. This setup allows the baseline to sample more candidates than SpSCO (which averages ~14.04 candidates per instance via its adaptive logic). Even under this setting, SpSCO achieves a better optimality gap (0.02%) compared to the multi-sample baseline (0.04%).
> >
> > SpSCO achieves this with superior efficiency, as it only triggers heavy sampling when necessary, whereas the baseline is forced to sample 16 times for every instance, regardless of difficulty. Similarly, in Table 2 for large-scale problems, we compare SpSCO against Prefix_Difusco (16 samples). SpSCO outperforms it on both TSP-500 (3.56% vs. 4.58%) and TSP-1000 (4.58% vs. 6.07%) while maintaining a faster inference time (e.g., 1.20m vs. 2.10m on TSP-500).
> >
> > To further rule out the advantage of simple sampling, we evaluated the POMO baseline with massive sampling rollouts (7,680 for TSP-500 and 3,840 for TSP-1000 shown in Table 3 in the paper draft) to strictly align with SpSCO's inference time. Even with this extensive sampling budget, the pure RL baseline failed to match SpSCO's solution quality. This confirms that SpSCO's advantage stems from the strategic guidance of the diffusion model at critical junctures, rather than merely increasing the quantity of unguided rollouts.
> >
> > Therefore, these results demonstrate that SpSCO's advantage stems from the quality of the RL-guided candidates rather than simply the quantity of rollouts.
> >
> > **Weakness 4:  Advantage of SpSCO Coming from Multiple Prefixes**
> >
> > **Response:**
> >
> > The reviewer correctly observes the standard sample-quality trade-off. However, we respectfully disagree that the benefit derives mainly from simple sampling. We argue that SpSCO fundamentally differs from brute-force sampling in both efficiency and mechanism.
> >
> > From the performance comparison between SpSCO and DM with the same number of rollouts, which we provide in the response to W3, SpSCO achieves better performance and faster inference time. These performance increase prove that the improvement is driven by the methodology of SpSCO, not raw sampling volume.
> >
> > **Fundamentally, the efficiency advantage stems from the synergy between the uncertainty trigger and the conditional sampling.** The **trigger** acts as a critical gatekeeper: it identifies the precise moment the greedy RL agent becomes uncertain, locking in the high-confidence portion of the tour as a robust prefix. Consequently, the diffusion model does not solve from scratch (starting from pure noise like standard sampling), but is **strategically deployed** to resolve only the complex sub-problems where the RL policy struggles. This "strategic speculation" serves as a guided search, allowing SpSCO to allocate its computational budget selectively—focusing samples only on the uncertain segments—whereas standard brute-force sampling wastes compute resolving the entire sequence complexity, including easy parts that the RL agent could have handled instantly.
> >
> > We have expanded Section 4.4 to explicitly highlight the adaptive nature of the candidate selection process to distinguish it from fixed-batch sampling.
> >
> > **Weakness 5:  Generalizability on Other CO Problems**
> >
> > **Response:**
> >
> > We have expanded our experiments to the maximum independent set (MIS) problem (in Appendix B.2) and added another OP problem with different scales, i.e., OP-50/200 (in Appendix B.1). All the results and analysis are stated in the global response. The results of MIS also show the strength of the SpSCO framework, consistent with our findings in the routing problems.

---

> ### Author Response · Authors · 2025-11-24
> **Response to Question 1, 2 and 3**
>
> **Q1**: Please clarify and comment on the issues raised under weaknesses.
>
> **Response:**
>
> Please see our response from W1 to W5.
>
> **Q2:** Can you provide ablation on the single-trigger design vs. allowing multiple triggers?
>
> **Response:**
>
> Please refer to the part of the global response named 'Practical and theoretical rationale of the single-trigger design'. We have provided a detailed ablation study comparing single-trigger and multiple-trigger designs there. The results demonstrate that allowing multiple triggers yields only marginal performance gains (e.g., $\sim$0.01% on TSP-100) but increases inference time by approximately 10-20 times, confirming that the single-trigger design offers a superior speed-quality trade-off.
>
> **Q3:** Why is the Attention Model slower than SpSCO? In my understanding, this cannot be the case, since the AM only invokes a single, greedy rollout with a similar architecture as the draft model of SpSCO, but without the dual trigger mechanism and DM generation.
>
> **Response:**
>
> This is a typo in reporting the inference time of the RL model; the accurate inference times per instance for the TSP500 and TSP1000 problems are 0.468s and 1.09s, respectively. We have corrected them accordingly. Thanks for spotting this typo.

---

### Official Review · Reviewer_nQiK · 2025-11-01

**Soundness:** 3
**Presentation:** 3
**Contribution:** 3
**Rating:** 6
**Confidence:** 5

**Summary:**

This paper proposes SpSCO, a speculative sampling framework inspired by large language models (LLMs). The framework utilizes a non-autoregressive reinforcement learning (RL) model, such as AM or POMO, as the draft model to generate initial solutions. When there is a significant discrepancy between the RL policy and the diffusion model (DM) policy, or when the RL policy behaves in a highly stochastic manner, the DM (serving as the target model) is activated to generate an alternative solution. The final output is selected as the better of the two. The proposed method is empirically evaluated on TSP instances of sizes 50, 100, 500, and 1000.

**Strengths:**

The idea of using a fast RL model as a draft generator and a high-quality but slower diffusion model (DM) as compensation is interesting. I agree with the underlying motivation: single-paradigm methods often suffer from inherent limitations. For instance, autoregressive RL models are typically efficient but yield only moderate solution quality, while non-autoregressive diffusion models can achieve superior performance at the cost of prolonged denoising procedures. Combining these two paradigms in a complementary manner is a promising direction for improving the trade-off between solution quality and inference efficiency in neural combinatorial optimization.

**Weaknesses:**

- The influence of different threshold selections should be discussed.

- Due to the reliance on a diffusion model, the proposed method also faces limitations in its applicability to more complex problems.

- The pseudo code is unnecessarily lengthy. Beyond this, the overall presentation requires further improvement for clarity and conciseness.

- The criteria for hyperparameter selection should be explained in Appendix A.4, rather than merely listing their values.

**Questions:**

- How frequently will the DM inference be triggered?

- How does the performance of LKH-3 in Table 2 compare to SpSCO under a shorter time limit?

- Why is 2-opt not used in large-scale TSP in Table 2?

- Why does AM trained on TSP-50/100 use rollout as the baseline, rather than the critic baseline used in the original AM?

- How is the performance on TSPLIB?

---

> ### Author Response · Authors · 2025-11-24
> **Response to Weakness 1, 2, 3, and 4**
>
> **W1: Influence of Different Thresholds**
>
> **Response:**
>
> We thank the reviewer for this suggestion. The entropy threshold serves as a sensitivity knob balancing solution quality and computational cost.
>
> 1. Findings on TSP:
>
> As detailed in Appendix B.4 Tables 14 and 15, lowering the threshold on TSP-100 increases trigger frequency, improving the optimality gap (e.g., from 0.09% to 0.01%) at the cost of inference time. Conversely, high thresholds suppress the trigger, reverting performance to the RL baseline.
>
> 2. Extended Analysis on Orienteering Problem (OP):
>
> We conducted a sensitivity sweep on the Entropy Threshold across OP-50, OP-100, and OP-200 with the KL threshold fixed, where the results are shown in the following table. The results reveal two key trends:
> * Performance Correlates with Trigger Rate: An optimal spot exists (e.g., 1.5 for OP-50/100). Setting the Entropy threshold too high (e.g., 3.0) drastically reduces the trigger rate (<10%), causing the average reward to drop significantly as the model fails to intervene.
> * Scale Adaptation: The optimal threshold shifts upwards for larger problems (from 1.5 for OP-50 to 2.5 for OP-200). This adjustment accounts for the naturally higher entropy inherent in larger action spaces.
>
> | Dataset | Entropy Threshold | Avg. Reward $\uparrow$ | Trigger Rate | Avg. Trigger Step |
> | :--- | :--- | :---: | :---: | :---: |
> | **OP-50** | 1.0 | 12.96 | 100.0% | 1.34 |
> | | 1.5 (**Best**) | **13.23** | 98.8% | 3.46 |
> | | 2.0 | 12.73 | 50.0% | 6.77 |
> | **OP-100** | 1.5 (**Best**) | **29.51** | 100.0% | 1.79 |
> | | 2.5 | 27.67 | 14.4% | 27.17 |
> | | 3.0 | 27.34 | 1.2% | 31.50 |
> | **OP-200** | 1.5 | 42.19 | 100% | N/A |
> | | 2.5 (**Best**) | **43.90** | 85.0% | 20.06 |
> | | 3.0 | 39.23 | 9.2% | 27.67 |
>
> **W2: Limitation Caused by the Diffusion Model Part**
>
> **Response:**
>
> We acknowledge this limitation. The applicability of SpSCO is indeed bounded by the capability of the underlying diffusion backbone, particularly for problems with highly complex constraints (e.g., VRP with complex time windows), where training effective diffusion models remains an open challenge in the field. It is worth noting that our hybrid design inherently mitigates this issue compared to pure diffusion solvers. Since SpSCO utilizes the RL agent for the actual step-by-step construction, we can strictly enforce hard constraints via validity masking within the RL decoder, utilizing the diffusion model primarily for probabilistic guidance rather than strict generation. Furthermore, as a model-agnostic framework, SpSCO is designed to be forward-compatible, allowing for the seamless integration of more advanced diffusion backbones as the broader research community continues to improve their capabilities on complex tasks.
>
> **W3: Long pseudo and Presentation**
>
> **Response:**
>
> We thank the reviewer for this constructive feedback. We have taken concrete steps to improve the manuscript's clarity and conciseness:
>
> 1. **Streamlining Algorithm 1:** We have significantly condensed the pseudo-code by abstracting the mathematical calculations of Entropy and Energy (Lines 5-11 in the original version) into the main text. Besides, we clarified the dual-track generation block and adjusted the pseudocode to a clearer and understandable one with added comments. The revised algorithm now focuses strictly on the high-level decision flow and the dual-track logic.
>
> 2. **Improving Clarity:** We revised Figure 2 to eliminate the visual contradiction regarding the "single-trigger" mechanism (on Page 5, Lines 216-238). Besides, we have rewritten Section 4.3 (Trigger Mechanism) and Section 4.4 (Dual-Track Generation) to eliminate ambiguities regarding the "Cognitive Divergence" definition and the "Hybrid Path" construction, addressing the specific clarity concerns raised by the reviewers.
>
> **W4: Criteria of Hyperparameter Selection in A.4**
>
> **Response:**
>
> We appreciate the reviewer's suggestion to clarify the rationale behind our hyperparameter choices. In the revised Appendix A.4 (Page 16), we have stated that our selection criteria are guided by two primary principles: fairness and efficiency. First, to ensure rigorous comparability, we strictly adopted the architectural configurations (e.g., layers, dimensions) from the original DIFUSCO and RL4CO implementations, thereby isolating the performance gains to the SpSCO framework logic rather than backbone scaling. Second, for inference-specific parameters such as DDIM sampling steps ($T_{inf}$), values were selected based on a preliminary analysis of the speed-quality trade-off; for instance, we minimized $T_{inf}$ (e.g., to 10) for small-scale problems to prioritize speed while increasing it (e.g., to 50) for larger scales to ensure robustness.

---

> > ### Author Response · Authors · 2025-11-24
> > **Response to Question 1, 2, 3, 4, and 5**
> >
> > **Q1:** How frequently will the DM inference be triggered?
> >
> > **Response:**
> >
> > Motivated by both practical and theoretical considerations, SpSCO is explicitly designed to trigger the diffusion model exactly once. For a detailed justification, please refer to the part of global response named ‘Practical and theoretical rationale of the single-trigger design’.
> >
> > Across different datasets of all problems, we observe a trigger participation rate of nearly 100%. This indicates that almost every instance eventually reaches a critical state where the greedy RL policy deviates from the global structural prior, which is consistent with the sub-optimal nature of the solutions constructed by the RL’s policy.
> >
> > **Q2:** How does the performance of LKH-3 in Table 2 compare to SpSCO under a shorter time limit?
> >
> > **Response:**
> >
> > We thank the reviewer for this interesting question. To facilitate a direct comparison, we conducted additional experiments on LKH-3 by strictly limiting its solving time to align exactly with SpSCO's inference time reported in Table 2 (i.e., 1.2 minutes for TSP-500 and 2.4 minutes for TSP-1000 per instance).
> >
> > On TSP-500 (Limit: 1.2m): LKH-3 achieves an average length of 17.81, whereas SpSCO achieves a superior 17.24. This means LKH-3 performs 3.31% worse (produces longer tours) than SpSCO under the same time constraint.
> >
> > On TSP-1000 (Limit: 2.4m): LKH-3 achieves an average length of 24.82, whereas SpSCO achieves 24.61. LKH-3 performs 0.85% worse than SpSCO.
> >
> > **Q3:** Why is 2-opt not used in large-scale TSP in Table 2?
> >
> > **Response:**
> >
> > We explicitly excluded 2-opt in the large-scale experiments to ensure a **fair and direct evaluation of the model's intrinsic generation capability**. Since 2-opt is a heuristic local search operator, applying it would mask the actual quality of the solutions constructed by the backbone model.
> >
> > Furthermore, we focused on the **trade-off between solution quality and inference latency** in the large-scale setting. Introducing 2-opt complicates this analysis because its runtime is highly dependent on the initial solution quality (i.e., better initial solutions may require fewer steps to converge, or conversely, trap the search in complex local optima), thus introducing significant variance that makes standardized runtime comparisons unreliable.
> >
> > **Q4:** Why does AM trained on TSP-50/100 use rollout as the baseline, rather than the critic baseline used in the original AM?
> >
> > **Response:**
> >
> > This is a typo of reporting the inference time of the RL model; the accurate inference time per instance for TSP-500 and TSP-1000 problems is 0.468s and 1.09s, respectively. We have corrected it in the revised version.
> >
> > **Q5:** How is the performance on TSPLIB?
> >
> > **Response:**
> >
> > We thank the reviewer for this suggestion. We have conducted a comprehensive evaluation on the TSPLIB benchmark and added the detailed instance-wise results and analysis to Appendix B.3 in the revised manuscript (Page 19). SpSCO achieves an average optimality gap of 1.90% across these instances.

---

> > > ### Comment · Reviewer_nQiK · 2025-11-28
> > >
> > > Thank you for the detailed response and the corresponding revision of the paper.
> > >
> > > The discussion on different entropies is interesting.
> > >
> > > I am not convinced by the response to Q3. I agree with the authors that "applying 2opt would mask the actual quality of the solutions constructed by the backbone model" (which I believe is a weakness of the original diffusion-based method), but I am confused why the authors hold inconsistent views across different problem sizes.
> > >
> > > Please check the response to Q4, which appears unrelated to my question.
> > >
> > > Although I appreciate the motivation of synergizing the RL-based method and the diffusion-based method, I feel that the current design of using the diffusion-based method as a fallback for the RL-based method is not elegant. The authors should consider why they need to be integrated and how to integrate them in a more organic way.
> > >
> > > I truly appreciate your efforts and hope you can dig deeper. I'm inclined to maintain the current score.

---

> ### Author Response · Authors · 2025-12-01
> **2-opt Results and Response to Q4**
>
> > Response to Q3 (2-opt): I am not convinced by the response to Q3.
>
> **Response:**
>
> About the 2-opt experiment, we have added the experiment results to the revised PDF. The detailed results are shown in the following table (LEHD is an autoregressive algorithm, which can be trained by the RL or SL method):
>
> | ALGORITHM | TYPE | TSP-500 LENGTH↓ | TSP-500 DROP↓ | TSP-500 TIME | TSP-1000 LENGTH↓ | TSP-1000 DROP↓ | TSP-1000 TIME |
> | :--- | :--- | :--- | :--- | :--- | :--- | :--- | :--- |
> | POMO | RL+G+2OPT | 17.77 | 7.37% | 0.58s | 25.10 | 6.67% | 1.05s |
> | LEHD | RL/SL+G+2OPT | 16.77 | 1.40% | 1.83s | 24.06 | 2.29% | 3.85s |
> | DIMES | RL+G+2OPT | 17.65 | 6.62% | 1.01m | 24.83 | 7.38% | 2.29m |
> | DIMES | RL+AS+G+2OPT | 17.31 | 4.57% | 2.10h | 24.33 | 5.22% | 4.49h |
> | DIFUSCO | SL+G+2OPT | 16.83 | 1.68% | 5.75m | 23.92 | 1.66% | 17.52m |
> | T2T | SL+G+2OPT | 16.86 | 1.92% | 2.42m | 23.86 | 1.42% | 15.90m |
> | SpSCO(POMO) | RL+SL+G+2OPT | 16.82 | 1.63% | 2.32m | 23.92 | 1.66% | 4.40m |
> | SpSCO(LEHD) | RL+SL+G+2OPT | **16.74** | ***1.20%*** | 0.98m | **23.64** | ***0.46%*** | 0.36m |
>
> Compared to traditional RL baselines such as POMO and DIMES, SpSCO demonstrates significantly superior performance in terms of optimality gap. Moreover, when contrasted with pure Diffusion Model (DM) algorithms, SpSCO not only outperforms them in solution quality but also exhibits a marked advantage in computational efficiency. These results highlight SpSCO's ability to achieve fast, robust, and high-quality inference. Crucially, these empirical findings validate the theoretical analysis of marginal regret dynamics illustrated in Figure 3. By strategically switching from the RL policy to the Diffusion Model at the optimal trigger point $t^*$, **SpSCO effectively leverages the strengths of both paradigms, yielding a lower cumulative regret than either backbone could achieve in isolation while accelerating traditional DM's inference time**.
>
> > Please check the response to Q4, which appears unrelated to my question.
>
> **Response:**
>
> We apologize for the confusion in our previous reply. We chose the Rollout Baseline for TSP-50/100 primarily to align with the state-of-the-art results reported in the original AM paper [1], where it was shown to act as a stronger baseline than a learned Critic. Empirically, the greedy rollout provides a robust low-variance baseline for these sizes. Moreover, since the baseline policy is frozen within each epoch, it operates in inference mode without gradient tracking, significantly reducing memory consumption. This efficiency enables the use of larger batch sizes, which accelerates training throughput.
>
> However, as the problem scale increases to TSP-500/1000, we observed that the Rollout method leads to severe training instability and convergence failure. In these high-dimensional search spaces, the "frozen" greedy policy becomes increasingly difficult for the stochastic policy to outperform, leading to infrequent updates and stale baseline estimates that exacerbate the variance of the REINFORCE estimator. Consequently, for large-scale instances, we transitioned to a Critic Baseline.
>
> [1] Kool, W., van Hoof, H., & Welling, M (2018). Attention, Learn to Solve Routing Problems!. In ICLR.

---

> ### Author Response · Authors · 2025-12-01
> **SpSCO Design Philosophy: A Theoretically Grounded Speculative Framework, Not a Heuristic Fallback**
>
> ## Summary
>
> We appreciate the reviewer's feedback on the integration design. We respectfully clarify that SpSCO is not designed as a heuristic "fallback" strategy for RL. On the contrary, our work **takes Diffusion Models (DMs) as the starting point** and utilizes a Speculative Sampling framework integrated with RL to unlock their practical potential. Furthermore, this is not a naive application; we propose a mathematically grounded adaptation that organically aligns the two models based on their distinct error dynamics. This design ensures the handoff strategy yields **a cumulative trajectory error strictly lower than that of either standalone backbone** while **accelerating DM’s inference time**, rather than serving as a simple heuristic patch.
>
> ## Why do RL and DM need to be integrated?
>
> As highlighted in our abstract, current neural CO solvers face a distinct speed-quality dilemma. Sequential RL solvers are **efficient** but suffer from **sequential error accumulation**, whereas DMs capture global structure for **high-quality solutions** but incur **prohibitive computational costs**. Integrating them is necessary to synergize the inference speed of RL with the generative quality of DMs, creating a solver that **transcends the limitations of either individual paradigm**.
>
> ## How are RL and DM integrated organically?
>
> Far from a rigid fallback, the integration is fundamentally grounded in our theoretical analysis (Appendix C). We first analytically establish that the RL agent’s marginal regret increases monotonically with the time step (specifically scaling as $\propto \sqrt{t}$)—typically driven by compounding variances and distributional shifts—whereas the DM’s marginal error remains constant. This theoretical framework guarantees the existence of **a unique, optimal switching point $t^*$** where the cumulative trajectory error is minimized (strictly lower than either standalone baseline, as shown in Figure 3).
>
> To capture this theoretical optimum in practice, SpSCO employs a dual-signal trigger (Policy Entropy and KL Divergence) designed to dynamically detect $t^*$ approximately. These signals are selected as specific, observable proxies for the latent error sources identified in our theory: RL’s high entropy signals the **high local variance** that drives compounding error, while high KL divergence explicitly signals the **distributional shift** away from the DM’s global solution manifold. By monitoring these signals, SpSCO identifies precisely when the RL policy becomes less reliable than the DM, rendering the handoff strategy theoretically grounded rather than heuristic.
>
> ## Experimental Support
>
> This design is empirically validated on large-scale benchmarks (TSP-500 and TSP-1000). Consistent with our theoretical analysis that the hybrid strategy minimizes cumulative regret, SpSCO achieves optimality gaps of 3.56% (TSP-500) and 4.58% (TSP-1000), which strictly outperform both the constituent RL backbone (**> 11.43\%  gap** on average) and the DM backbone (**> 4.06% gap** ). Similarly, SpSCO also outperforms two backbones in OP and MIS settings. Simultaneously, SpSCO enables practical deployment by significantly reducing inference latency; it solves instances in just 1.20 min and 2.43 min, respectively—**approximately $4\times$ faster** than standard Diffusion Models (e.g., DIFUSCO/T2T). In doing so, SpSCO establishes a robust solver that **effectively transcends the inherent limitations of singular RL and DM paradigms**.

---

### Official Review · Reviewer_rLv2 · 2025-11-01

**Soundness:** 3
**Presentation:** 3
**Contribution:** 3
**Rating:** 6
**Confidence:** 3

**Summary:**

This paper proposes SpSCO, a hybrid framework that combines reinforcement learning (RL) and diffusion models (DMs) using a speculative sampling (SpS) mechanism inspired by recent work in LLM inference acceleration. The RL solver serves as a lightweight draft model, while the DM acts as a high-capacity target model. A dual-signal adaptive trigger, based on (1) the RL policy entropy and (2) the KL divergence between the RL and DM-derived priors, determines when to invoke the DM. When triggered, the DM performs a one-step correction and generates a few complete solutions. A final selection step chooses the best overall tour.

Experiments on standard TSP benchmarks (50–1000) demonstrate strong results: SpSCO achieves near-optimal quality with substantially reduced inference time compared to pure diffusion-based methods.

**Strengths:**

- The idea of transferring speculative sampling from LLMs to combinatorial optimization is quite novel and timely.

- The dual-signal trigger (entropy + cognitive divergence) is intuitive and empirically effective.

- Experiments are extensive, and the results on large-scale TSPs show a promising trade-off between speed and quality.

- The method is described as model-agnostic, and the implementation details (especially for Prefix-Difusco) are very thorough, which supports reproducibility.

**Weaknesses:**

- The notion of “cognitive divergence” between RL and DM distributions is attractive, but it feels more metaphorical than grounded. The paper frames it as if the DM provides a “global understanding” of the solution space, but this is not rigorously demonstrated. In practice, the DM prior derived from a single-step denoising probe is only a local statistical snapshot, not an actual global consistency check.

- While TSP results are solid, there’s limited exploration of other CO problems—the single Orienteering example in the appendix is not enough to claim broad generality.

**Questions:**

- How expensive is the energy probing step compared to full DM sampling? what’s the overhead in practice?
- How does SpSCO perform if both RL and DM are fully trained, rather than under-trained?
- Could this idea extend to constrained problems like VRP or CVRP, where feasibility must be enforced?

---

> ### Author Response · Authors · 2025-11-24
> **Response to Weakness 1 and 2**
>
> **Weakness 1: Cognitive Divergence is metaphorical than grounded**
>
> **Response:**
>
> We appreciate the reviewer's critique of the theoretical grounding of our metric, but respectfully disagree. We believe that the "Cognitive Divergence" is firmly grounded in the global receptive mechanism of the diffusion model, rather than being a mere metaphor.
>
> First, we refute the notion that the single-step probe is only a "local statistical snapshot." The diffusion backbone in SpSCO is a Graph Neural Network (GNN) with 12 layers of message passing, a global receptive field, trained to predict the complete clean adjacency matrix ($\hat{x}\_0$) at every denoising step, not just local edges.  Mathematically, the energy score $E_{\phi}$ derived from this probe acts as a proxy for the negative log-likelihood of the candidate action conditioned on the entire graph topology. Thus, it does perform a global consistency check.
>
> Consequently, the term "Cognitive Divergence" rigorously describes the quantifiable disagreement between two distinct generative paradigms: the RL's sequential, myopic distribution ($\pi_{\theta}$) versus the DM's holistic, structure-aware global prior ($p_{\phi}$). The KL divergence ($D_{KL}$) serves as the information-theoretic metric to measure the deviation between these "local" and "global" views. The practical grounding of this metric is further validated by our ablation study (Table 11), where adding the KL trigger improves the optimality gap from 0.05% to 0.02%, proving it captures critical structural discrepancies that internal uncertainty alone misses.
>
> **Weakness 2: Generalizability on Other CO Problems**
>
> **Response:**
>
> Thanks for your comment. We have stated further experimental results and corresponding analysis of OP and MIS in the part of the global response, named ‘New results on other problem instances’. All the results reveal a consistent trend: SpSCO effectively synergizes the strengths of its components, outperforming other baselines across various benchmarks.

---

> ### Author Response · Authors · 2025-11-24
> **Response to Question 1, 2 and 3**
>
> **Q1**:How expensive is the energy probing step compared to full DM sampling? what’s the overhead in practice?
>
> **Response**:
>
> We record the computational time of the energy probing and full DM sampling (taking TSP100 and TSP500 as an example) as follows:
>
> The energy probing step is effectively cost-free in practice. The computational overhead is negligible (recorded as < 1ms, effectively negligible) compared to the massive computational cost of full DM sampling.
>
> **Q2**:  How does SpSCO perform if both RL and DM are fully trained, rather than under-trained?
>
> **Response**:
>
> We thank the reviewer for this question regarding the impact of backbone model convergence on SpSCO's performance.
>
> 1. TSP-50 and TSP-100: As noted in our implementation details, the models used for TSP-50 and TSP-100 in the main paper were trained for 50 epochs following the standard protocol in DIFUSCO (Sun et al., 2023). These represent the "fully trained" convergence state. Thus, the results reported in Table 1 of the main paper already reflect the performance of fully trained backbones.
>
> 2. Large-scale Instances (TSP-500/1000): For larger scales, training constraints are more significant. We conducted additional experiments to isolate the impact of training maturity for both the RL policy and the Diffusion Model (DM).
> Equipped with fully-trained DM, we evaluated SpSCO with varying degrees of RL backbone convergence (100 epochs vs. 350 epochs).
>
> We compared our default "under-trained" DM (20 epochs) against a "fully trained" DM (50 epochs), both equipped with the fully trained RL backbone. As shown in the following Table 1, extending the DM training yields consistent performance gains, reducing the optimality gap from 3.56% to 2.98% on TSP-500 and from 4.58% to 4.39% on TSP-1000.
>
> **Table 1: Performance of SpSCO with a mature RL policy (350 epochs) across different Diffusion Model training stages.**
> ***
>
> | Training epoch of the diffusion model | TSP-500 (Gap) | TSP-1000 (Gap) |
> | :---: | :---: | :---: |
> | 20 | 3.56% | 4.58% |
> | 50 | 2.98% | 4.39% |
>
> ***
>
> The following Table 2 demonstrates that a better-trained RL draft model significantly improves the final solution quality (e.g., reducing the gap from 7.07% to 4.58% on TSP-1000).
>
> **Table 2: Performance of SpSCO with a fully trained Diffusion Model (50 epochs) across different RL training stages.**
> ***
>
> | Training epoch of the RL model | TSP-500 (Gap) | TSP-1000 (Gap) |
> | :---: | :---: | :---: |
> | 100 | 4.26% | 7.07% |
> | 350 | 2.98% | 4.39% |
>
> ***
>
> Regarding the RL component, we observed that the validation performance plateaued and stabilized around 350 epochs, indicating that the RL policy had reached empirical convergence (maturity).
>
> Therefore, to evaluate SpSCO with both mature components, we combined this converged RL policy (350 epochs) with the fully trained Diffusion Model (50 epochs). Based on the experimental results shown in the tables above:
>
>
>
> * TSP-500: The combination of Mature RL and Mature DM reduces the optimality gap to 2.98%.
> * TSP-1000: This combination achieves an optimality gap of 4.39%.
>
> Please note that the result for TSP-500(3.56%) and TSP1000(4.58%)  in the previous main paper was obtained using the Mature RL coupled with the Under-trained DM. Upgrading the DM to its mature state yields further improvements reported here.)
>
> While the main paper highlights that SpSCO achieves comparable results even with suboptimal, under-trained backbones (demonstrating robustness), providing fully trained backbones further pushes the performance boundary, yielding even lower optimality gaps. However, this also brought about a several-fold increase in the training load of each backbone.
>
> **Q3**: Could this idea extend to constrained problems like VRP or CVRP, where feasibility must be enforced?
>
> **Response**:
>
> Please refer to the **"New results on other problem instances"** section in the Global Response. To demonstrate SpSCO's capability on constrained problems (such as VRP/CVRP), we specifically evaluated it on the Orienteering Problem (OP). OP imposes a strict maximum length constraint, requiring the solver to balance reward collection with feasibility, which shares the same challenge as VRP. As shown in the Global Response, SpSCO achieves superior performance on OP, confirming its effectiveness in constrained scenarios.

---

### Official Review · Reviewer_CJH2 · 2025-11-01

**Soundness:** 2
**Presentation:** 1
**Contribution:** 1
**Rating:** 2
**Confidence:** 4

**Summary:**

This paper introduces SpSCO, a hybrid framework for neural combinatorial optimization that addresses the speed-quality trade-off between fast autoregressive models (trained with RL) and slow, high-quality diffusion models (trained with SL). Inspired by speculative sampling in language generation, SpSCO uses the RL model to draft a solution and adaptively trigger the diffusion model for correction only at a "critical" step. This critical step is identified by the entropy of the RL policy or its KL divergence from the diffusion model prior. The proposed method is tested on TSP with various scales.

**Strengths:**

1. Adopting the speculative decoding idea to neural combinatorial optimisation is an interesting idea.
2. Combining autoregressive models and a diffusion-based heatmap generation model is a promising research direction to explore, since they have different pros and cons, as mentioned in this work.

**Weaknesses:**

1. (Clarity) This work has serious concerns in terms of clarity. Most importantly, "RL model" is not in the same category as "diffusion models" since RL is a learning algorithm while diffusion models are a form of generative model. It should be fixed to "autoregressive models", which they used in their experiments. Equation (5) and (6) are hard to understand, and need more explanation, e.g., how $c_i$ and $t_{\text{probe}}$ are defined or obtained. $p_\phi$ was defined as a diffusion kernel in equation (2), but used differently in (6). Algorithm 1 seems to have various errors, e.g., once $dm_triggered$, the $a_{next}$ is never sampled again, and hybrid path correction is missing. See my questions for other points regarding the clarity.
2. (Method) To me, some of the design seems heuristic. For example, the trigger is activated only once, i.e., the proposed speculative decoding is applied only once. Moreover, the diffusion models seem to be assumed to be trained via supervised learning, which limits the "data-free" nature of RL-based methods.
3. (Experiments) The experiments have a limited scope, in that the method is validated only on TSP. While it is claimed to achieve the state-of-the-art performance, it is actually far from it as far as I know; see the following works [1, 2, 3, 4]. No standard deviation or confidence interval is provided. The runtime for Table 1 is also not provided.

---
References
[1] Luo, Fu, et al. "Neural combinatorial optimization with heavy decoder: Toward large scale generalization." Advances in Neural Information Processing Systems 36 (2023)
[2] Drakulic, Darko, et al. "Bq-nco: Bisimulation quotienting for efficient neural combinatorial optimization." Advances in Neural Information Processing Systems 36 (2023)
[3] Kim, Minsu, et al. "Ant Colony Sampling with GFlowNets for Combinatorial Optimization." The 28th International Conference on Artificial Intelligence and Statistics. (2025)
[4] Kim, Hyeonah, et al. "Neural Genetic Search in Discrete Spaces." Forty-second International Conference on Machine Learning. (2025)

**Questions:**

1. Line 144: Why are diffusion models better than other models (like autoregressive models) in capturing complex global dependencies?
2. Line 235: "the RL policy completes the remainder...": If I understand correctly, the remainder is completed via both the RL policy and the diffusion model, isn't it?
3. Line 443: Why is AM slower than SpSCO?
4. Table 2: Is 2OPT used for the large-scale experiment?

---

### LLM usage disclosure
I used LLM only to check grammar.

---

> ### Author Response · Authors · 2025-11-24
> **Response to Weakness 1 and 2**
>
> **Weakness 1: Clarifications**
>
> **Response:**
>
> Thanks for the Good point. We do agree that RL and diffusion models are not in the same category, and we clarified the corresponding sentences that caused your misunderstanding. We have made the following changes in the updated PDF.
>
> * We clarified Equations 5 and 6 on page 6, Lines 305-323.
>
> * For Equations (2) and (6), we used the same mathematical symbols to represent different distributions in the original version, which led to your confusion. We change Equation (6) to a different symbol $q_{DM}$,  to avoid any misunderstandings on Page 6, Line 321.
> * We largely changed the algorithm for better clarity and conciseness. The major changes include: We clarified the dual-track generation block and adjusted the pseudocode to a clearer and understandable one with added comments. To show hyperparameters explicitly, we added missing inputs, specifically the Top-$P$ threshold ($P_{thresh}$) and Probe size ($M$), to clarify the sampling parameters. We also added explicit references to equations within the algorithmic steps. This allows readers to strictly trace how entropy, energy, and the DM prior are calculated. Furthermore, we updated the notation to use $q_{\text{DM}}$ for the KL divergence calculation, ensuring consistency with the updated text.
>
> **Weakness 2: Trigger Design and Data Training**
>
> **Response:**
>
> We thank the reviewer for these insightful comments. About the design of the trigger, especially why the trigger is activated once, we provide detailed justifications for both the **practical implications and theoretical insights** of the SpSCO framework in the global response, named ‘Practical and theoretical rationale of the single-trigger design’.
>
> Besides, we fully agree with the reviewer’s observation regarding the data nature of our method. However, we clarify that our primary objective is to propose a **hybrid framework** that optimizes the **trade-off between solution quality and inference time**, rather than solely improving RL in its original paradigm. Hence, the ‘Data-free’ property is not the focus of our work.

---

> ### Author Response · Authors · 2025-11-24
> **Response to Weakness 3**
>
> **Weakness 3: Experimental Results**
>
> **Response:**
>
> 1. **On SpSCO’s Generalizability (OP & MIS)**
>
> We appreciate the suggestion to expand the scope and have significantly expanded our experimental scope to demonstrate the generalizability of SpSCO. The concrete experimental results and analysis about OP and MIS are explained in the global response, named ‘New results on other problem instances’.
>
> 2. **Correction of "State-of-the-Art" Claim and Comparison with Baselines**
>
> We sincerely appreciate the reviewer’s valuable correction regarding our "State-of-the-Art" (SOTA) statement. We agree that our original claim was not accurate, given the diverse landscape of NCO solvers.
>
> In the revised PDF, we have explicitly corrected this statement as follows (in the Abstract):
> “It also shows strong robustness: even with under-trained, suboptimal RL and diffusion backbones, SpSCO achieves a better trade-off between performance and inference time compared with strong RL and DM backbones on diverse CO instances across various scales while attaining faster inference time on large-scale instances.” We also thank the reviewer for sharing the related works ([1]-[4]). We have added a detailed discussion of these papers in the Related Work section (page 2 os Section5.1) for a more comprehensive overview of the current NCO landscape and to better clarify SpSCO's position.
>
> * **Iterative Meta-Heuristics** ([3] ACS-GFN, [4] Neural Genetic Search): These methods integrate neural networks with heavy iterative search algorithms (Ant Colony, Genetic Algorithms). While they achieve impressive solution quality, they operate as **search-based improvements**. They typically require significantly longer runtimes (minutes to hours) compared to SpSCO’s efficient inference (~1 second for TSP-100).
> * **Heavy Decoder** [1]: As the name implies, this method relies on a heavy decoding process that trades significant computational cost for quality.
> * **Graph Reduction & Generalization** ([2] BQ-NCO): BQ-NCO leverages graph coarsening primarily to enable zero-shot generalization to massive scales. While efficient, it inherently trades off structural resolution for scalability. In contrast, SpSCO focuses on a **different dimension of the speed-quality trade-off**: it utilizes the generative precision of diffusion models to pursue the ultimate optimality (e.g., 0.02% gap) on standard benchmarks, while significantly reducing the inference cost (time) compared to standard diffusion solvers via speculative sampling.
>
> In summary, SpSCO fills a critical gap in the current NCO literature. Unlike the search-intensive methods [1, 3, 4] or coarsening methods [2], SpSCO is positioned as a high-efficiency constructive solver. It uniquely balances the trade-off by retaining the speed of RL draft models while leveraging the generative precision of diffusion models only at critical junctures.
>
> 3. **Missing Statistics: Standard Deviation and Runtime in Table 1**
>
> We thank the reviewer for this point. We have verified the stability of our method by rerunning TSP-50, TSP-100, TSP-500, and TSP-1000 experiments with five different random seeds. The Inter-Seed Standard Deviation (STD) is 0.00%. for TSP-50, TSP-100 and TSP-500，STD is 0.02% for TSP-1000. We also added the Instance-Level STD in the revision to reflect the performance variance across problem instances (e.g.,± 0.09% for TSP-50, ±0.05% for TSP-100,  ±1.91for TSP-500, ± 0.98% for TSP-1000).
>
> About the Runtime for Table 1, the average inference times are approximately 0.1s for TSP-50 and 1.2s for TSP-100 per instance on an NVIDIA A40. We did not include these specific values in Table 1 because both fall within a **highly efficient, low-latency regime** (seconds or sub-seconds). At this small scale, the runtime difference is largely dominated by the **fixed computational overhead** of the deep learning framework and the trigger mechanism, rather than exponential algorithmic complexity.
>
> **References**:
>
> [1] Luo, Fu, et al. "Neural combinatorial optimization with heavy decoder: Toward large scale generalization." Advances in Neural Information Processing Systems 36 (2023)
>
>  [2] Drakulic, Darko, et al. "Bq-nco: Bisimulation quotienting for efficient neural combinatorial optimization." Advances in Neural Information Processing Systems 36 (2023)
>
> [3] Kim, Minsu, et al. "Ant Colony Sampling with GFlowNets for Combinatorial Optimization." The 28th International Conference on Artificial Intelligence and Statistics. (2025)
>
>  [4] Kim, Hyeonah, et al. "Neural Genetic Search in Discrete Spaces." Forty-second International Conference on Machine Learning. (2025)

---

> ### Author Response · Authors · 2025-11-24
> **Response to Question 1, 2, 3 and 4.**
>
> **Q1:**
> Line 144: Why are diffusion models better than other models (like autoregressive models) in capturing complex global dependencies?
>
> **Response**:
>
> While autoregressive models encode global context, their sequential decoding forces early decisions to be fixed without foresight, making it difficult to coordinate distant parts of the solution. In contrast, diffusion models generate the solution holistically by refining the entire adjacency matrix simultaneously. This allows them to model the joint distribution of all edges, capturing complex global dependencies and topological structures that are often lost in sequential processes.
>
> **Q2:**
> Line 235: "the RL policy completes the remainder...": If I understand correctly, the remainder is completed via both the RL policy and the diffusion model, isn't it?
>
> **Response**:
>
> We appreciate the opportunity to clarify the specific mechanics of the "Hybrid Path" construction. The reviewer is partially correct that the overall framework utilizes both models, but specifically for the Hybrid Path correction track referenced in Line 235, the RL policy indeed completes the remainder independently after the single DM intervention.
> To be precise, once the trigger fires at step $k^\*$, for the hybrid path in the dual-track strategy, the DM is queried once to provide a corrected next action $a_{k^*}^{DM}$ for RL. After appending this single corrected step, the RL policy takes over exclusively to autoregressively complete the rest of the tour without further DM checks. This design ensures the hybrid path benefits from a critical correction while maintaining inference speed.
> We have revised the description in Section 4.4 (page 7) and Algorithm 1 (page 8) to explicitly distinguish between the "RL-completed Hybrid Path" and the "Full DM Proposals" to prevent this confusion.
>
> **Q3:**
> Line 443: Why is AM slower than SpSCO?
>
> **Response**:
>
> This is a typo of reporting the inference time of the RL model; the accurate inference time per instance for tsp500 and tsp1000 problems is 0.468s and 1.09s, respectively.
>
> **Q4:**
> Table 2: Is 2OPT used for the large-scale experiment?
>
> **Response**:
>
> We explicitly excluded 2-opt in the large-scale experiments to ensure a **fair and direct evaluation of the model's intrinsic generation capability**. Since 2-opt is a heuristic local search operator, applying it would mask the actual quality of the solutions constructed by the backbone model.
>
> Furthermore, we focused on the **trade-off between solution quality and inference latency** in the large-scale setting. Introducing 2-opt complicates this analysis because its runtime is highly dependent on the initial solution quality (i.e., better initial solutions may require fewer steps to converge, or conversely, trap the search in complex local optima), thus introducing significant variance that makes standardized runtime comparisons unreliable.

---

> > ### Comment · Reviewer_CJH2 · 2025-11-24
> >
> > I appreciate your endeavor in rebuttal. Here are some additional comments.
> >
> >
> > > About the design of the trigger, especially why the trigger is activated once, we provide detailed justifications for both the practical implications and theoretical insights of the SpSCO framework in the global response, named ‘Practical and theoretical rationale of the single-trigger design’.
> >
> > I skimmed the Theorem and the proof, and I'm unsure whether the assumptions are reasonable. 1) How to define the target distribution $q_{\delta}$ (shouldn't this be dirac-delta?), and how reasonable is it to assume that it is approximable by diagonal Gaussians? 2) The regret for RL seems to be defined with the optimality gap, while the regret for diffusion is defined as the cost that measures reconstruction error. I'm not sure how valid it is to compare two different regrets to theoretically justify the algorithmic design.
> >
> >
> > > However, we clarify that our primary objective is to propose a hybrid framework that optimizes the trade-off between solution quality and inference time, rather than solely improving RL in its original paradigm.
> >
> > I understand the focus of your work, but my concern is that relying on SL significantly undermines the algorithm's practicality. I believe one of the key motivations of the RL-based NCO solver is to replace the heuristic solver that requires domain expertise and extensive tuning, so that it can solve new problems with real-world constraints. Since SpSCO relies on the SL model, it eliminates this unique practicality of RL.
> >
> > > While they achieve impressive solution quality, they operate as search-based improvements. They typically require significantly longer runtimes (minutes to hours) compared to SpSCO’s efficient inference (~1 second for TSP-100).
> >
> > > Heavy Decoder [1]: As the name implies, this method relies on a heavy decoding process that trades significant computational cost for quality.
> >
> > When comparing results for TSP500 from [1, 2], I found that both LEHD [1] greedy decoding and NGS [2] show better solution quality and faster speed than SpSCO. Note that NGS [2] is based on RL training and doesn't need the (near-)optimal solutions to train the models.
> >
> > ---
> >
> > Since my main concerns are not entirely resolved, I will keep my score.
> >
> > ---
> > ### References
> > [1] Luo, Fu, et al. "Neural combinatorial optimization with heavy decoder: Toward large scale generalization." Advances in Neural Information Processing Systems 36 (2023)
> > [2] Kim, Hyeonah, et al. "Neural Genetic Search in Discrete Spaces." Forty-second International Conference on Machine Learning. (2025)

---

> > > ### Author Response · Authors · 2025-11-27
> > > **Responses to Questions 1 and 2**
> > >
> > > > skimmed the Theorem and the proof, and I'm unsure whether the assumptions are reasonable. 1) How to define the target distribution $q_{\delta}$ (shouldn't this be dirac-delta?), and how reasonable is it to assume that it is approximable by diagonal Gaussians? 2) The regret for RL seems to be defined with the optimality gap, while the regret for diffusion is defined as the cost that measures reconstruction error. I'm not sure how valid it is to compare two different regrets to theoretically justify the algorithmic design.
> > >
> > > **Response:**
> > >
> > > 1\) '**How to define the target distribution?**'
> > >
> > > $q_{\delta}$ refers to the approximate target distribution defined by the "early stopping time" $\delta$, where $q_0 = p_{data}$. As discussed in [1] (Section 1.1), adopting an early stopping time $\delta$ is standard in Diffusion Models because the score function $\nabla \log q_t$ tends to diverge as $t \to 0$ for non-smooth data distributions. Besides, the small distance caused by $\delta$, such as the Wasserstein-p metric between $q_{\delta}$ and $p_{data}$, is acceptable. In the DM mathematical theory, with the time $t \rightarrow \infty$, the distribution almost certainly converges to a sample-centered Dirac-delta function that the reviewer proposed. However, it describes the extreme behavior of a random process rather than a marginal distribution within a finite time step that most DM use.
> > >
> > > [1] Montanari, A. (2023). Nearly d-linear convergence bounds for diffusion models via stochastic localization. ICLR.
> > >
> > > '**How reasonable is it to assume that it is approximable by diagonal Gaussians?**'
> > >
> > > First, we are only assuming a Gaussian distribution in the **local area, not the entire solution space**. Second, the diagonal Gaussian can actually be relaxed to a **multivariate Gaussian with bounded eigenvalues**, and the regret analysis still holds. We have changed Theorem 2,  the corresponding proof, and Remark 2 about CO examples (Page 24). See our updated PDF for details.
> > >
> > > 2\) '**the regret for diffusion is defined as the cost that measures reconstruction error**'
> > >
> > > We believe that it is a misunderstanding that the regret for diffusion is defined as the cost that measures reconstruction error. The cost function defined in the theorem is the objective function $f$ that we usually use in the optimization area. We have changed ‘the cost function $C’$ into ‘the objective function $f’$ in the revised PDF to avoid possible confusion. Therefore, the regret of diffusion is the expectation objective gap between two different data distributions, the target distribution $q_{\delta}$ and the actual sampling distribution $p_{t_N}$ at the final timestep $t_N$,  which has the same definition as RL’s regret.
> > >
> > > > I understand the focus of your work, but my concern is that relying on SL significantly undermines the algorithm's practicality. I believe one of the key motivations of the RL-based NCO solver is to replace the heuristic solver that requires domain expertise and extensive tuning, so that it can solve new problems with real-world constraints. Since SpSCO relies on the SL model, it eliminates this unique practicality of RL.
> > >
> > > **Response:**
> > >
> > > We respectfully disagree with the reviewer’s comment that ‘Since SpSCO relies on the SL model, it eliminates this unique practicality of RL’. We clarify that the "practicality" our work aims to address is specifically the **inference latency** bottleneck of generative solvers, NOT the data availability at **training time**.
> > >
> > > We would like to highlight that our starting point is to **improve DM-based neural solvers** – *If the reviewer’s statement is true, then this may render the entire line of DM-based neural combinatorial solvers impractical*. In this context, the dependency on SL is an inherent characteristic of the current high-performance generative baseline, not a new limitation introduced by SpSCO. The motivation for using RL-based approaches is to improve the test time trade-off between performance and efficiency, not the label-free training paradigm of pure RL.
> > >
> > > Therefore, SpSCO does not compromise the practicality of RL; rather, it **harnesses** the inference speed of RL to **unlock the practical deployment** of Diffusion Models. By bridging the gap between the superior solution quality of generative models and the efficiency requirements of real-world scenarios, SpSCO makes high-performance Diffusion Models a viable option for practical applications.

---

> ### Author Response · Authors · 2025-11-27
> **Response to Question 3**
>
> > When comparing results for TSP500 from [1, 2], I found that both LEHD [1] greedy decoding and NGS [2] show better solution quality and faster speed than SpSCO. Note that NGS [2] is based on RL training and doesn't need the (near-)optimal solutions to train the models.
>
> **Response:**
>
> We thank the reviewer for providing these detailed comparisons. We have carefully analyzed the results regarding NGS and LEHD and would like to clarify the positioning of SpSCO relative to these methods (The following results of LEHD and NGS are tested on our dataset in an A40 GPU).
>
> **Comparison with LEHD (SpSCO is Model-Agnostic):** We acknowledge LEHD as a strong SOTA baseline. However, strictly speaking, SpSCO is a model-agnostic inference framework, whereas LEHD is a specific model architecture. Since LEHD is also an autoregressive constructive model, it is **compatible with SpSCO as a stronger backbone**. This would likely significantly reduce the trigger rate of the diffusion verifier, thereby further improving SpSCO's efficiency and solution quality.
>
> To demonstrate this, we integrated LEHD as the draft model within SpSCO. As shown in the following table (last row), SpSCO (LEHD+Difusco) successfully improves the optimality gap of the original LEHD from **1.64%** to **1.57%**. In the meantime, the inference time drops from **1.20 min** (with POMO) to **43.82s** (with LEHD).
>
> **Comparison with NGS:** As shown in the following table, while NGS achieves a good optimality gap (2.06%), it requires 2.67 minutes per instance. Our new variant, SpSCO (LEHD+Difusco), **strictly dominates NGS** in both solution quality (**1.57%** vs. 2.06%) and speed (**43.82s** vs. 2.67 min), achieving around $4\times$ speedup.
>
> | Algorithms | Gap | Time/per instance |
> | :--- | :--- | :--- |
> | LEHD (SL) | 1.64% | 0.59s |
> | NGS (RL+Search) | 2.06% | 2.67m |
> | POMO (Pure RL) | 15.62% | **0.49s** |
> | Difusco (Prefix DM) | 8.23% | 0.50m |
> | SpSCO (POMO+Difusco) | 3.56% | 1.20m |
> | SpSCO (LEHD+Difusco) | **1.57%** | 43.82s |

---

### Author Response · Authors · 2025-11-24
**Global Response - Strengths & Update Summary**

Dear reviewers,

We thank you for your thoughtful and valuable feedback. We are encouraged to see your recognition of the following strengths of our paper:

* **Novel Motivation & Promising Direction:** All reviewers acknowledged the novelty of adapting speculative sampling to NCO. Reviewers CJH2, nQiK, and rLv2 highlighted that combining autoregressive and diffusion models is a "promising research direction" and "timely" for addressing the speed-quality trade-off.

* **Effective & Extensible Framework:** Reviewers rLv2 and sc5a praised the dual-signal trigger design as "conceptually elegant" and "intuitive". They further valued the framework's "model-agnostic" nature, which supports "plug-and-play" applications across different backbones.

* **Strong Empirical Validation:** Reviewers rLv2 and sc5a commended the "extensive experiments" and "comprehensive empirical evaluation" on varying TSP scales. The "thorough implementation details" and ablation studies were also recognized for supporting reproducibility and robustness.

**[TL;DR: major updates]** We rigorously addressed and incorporated the feedback to strengthen our work. Below are the main changes to the updated version. All major updates are highlighted in red font in the new PDF.

1. **Clarification of SpSCO:** Algorithm 1 (Page 7), trigger metric (Section 4.3.2, Page 6), and dual-track solution generation (Section 4.4, Page 6).

2. **New results on other problem instances to show SpSCO’s generalizability:** We have added further experimental results and corresponding analysis of Orienting Problem (OP) and Maximum Independent Set (MIS) in Appendix B.1 (Page 18) and B.2 (Page 19), respectively.

3. **Practical and theoretical rationale of the single-trigger design:** We provide a comprehensive illustration of the rationale for the single-trigger design from both the practical implications and theoretical analysis (in Section 4.2), summarized as follows.

---

> ### Author Response · Authors · 2025-11-24
> **Global Response - New results on other problem instances**
>
> We now provide detailed discussions regarding the above Points 2 (new results on other problem instances) and 3 (Practical and theoretical rationale of the single-trigger design). We will address the rest of each reviewer's specific comments in the individual responses, including Point 1 (Clarification of SpSCO) in TL;DR.
>
> ## New results on other problem instances
> As demonstrated in the following two tables, the performance of SpSCO on the orienteering problem (OP) is on average 3.54\% higher than that of the high-quality DM backbone with the same number of rollouts across different datasets. Similarly, it is on average 1.68\% higher on the maximum independent set (MIS) problem. This indicates that the SpSCO framework can achieve high solution quality on other CO problems, not just on TSP.
>
> Besides, we also compare the performance and runtime with diffusion model backbones that were sampled at a much higher rate than those of SpSCO. In OP, even with only 1/8 of the sample size ($S_{cand}=8$ vs. $s=64$), SpSCO consistently outperforms the high-sampling diffusion baseline across all OP benchmarks. Notably, SpSCO achieves an approximate 2x inference speedup on OP-50 and OP-100, and on OP-200, it reduces the optimality gap by over 60% (from 2.77% to 1.08%) with comparable runtime.
>
> Similarly, in MIS, even with less than 1/10 of the sample size ($S_{cand}=5$ vs. $s=64$), SpSCO consistently yields superior solution quality compared to the high-sampling diffusion baseline across all benchmarks. Notably, SpSCO achieves an approximate 1.7x inference speedup on RB-LARGE, while reducing the optimality gap by over 36% on ER-700-800 (from 9.03% to 5.76%) and over 55% on SATLIB (from 0.27% to 0.12%).
>
> **Table: Performance comparison across OP-50, OP-100, and OP-200 benchmarks.** Score denotes the average collected prize (higher is better). Gap indicates the percentage deviation from the optimal solution (lower is better). Time is the average inference time in seconds. S: Sample Decoding. $S_{cand}$ is the candidate number of full DM proposals $\mathcal{P}_{DM}$.
>
> ***
>
> | Method | OP-50 Score $\uparrow$ | OP-50 Drop $\downarrow$ | OP-50 Time $\downarrow$ | OP-100 Score $\uparrow$ | OP-100 Drop $\downarrow$ | OP-100 Time $\downarrow$ | OP-200 Score $\uparrow$ | OP-200 Drop $\downarrow$ | OP-200 Time $\downarrow$ |
> | :--- | :---: | :---: | :---: | :---: | :---: | :---: | :---: | :---: | :---: |
> | Gurobi-300 | 14.37 | 0.00% | 300.0s | 32.10 | 0.00% | 300.0s | 44.38 | 0% | 300.0s |
> | Gurobi-30 | 14.31 | 0.40% | 30.0s | 31.13 | 3.02% | 30.0s | 31.37 | 29.31% | 30.0s |
> | AM (RL) | 11.92 | 17.04% | **0.03s** | 28.37 | 11.62% | **0.03s** | 38.59 | 13.05% | **0.09s** |
> | prefix_Coexpander (DM, s=1) | 12.27 | 14.61% | 0.09s | 28.64 | 10.77% | 0.15s | 41.36 | 6.80% | 0.19s |
> | prefix_Coexpander(s=8) | 12.42 | 14.01% | 0.12s | 29.05 | 9.49% | 0.21s | 42.54 | 4.15% | 0.47s |
> | Coexpander(s=64) | 12.51 | 12.94% | 1.08s | 29.20 | 9.03% | 0.89s | 43.15 | 2.77% | 2.67s |
> | **SpSCO($S_{cand}$=8)** | **13.23** | **7.93%** | 0.55s | **29.51** | **8.06%** | 0.41s | **43.90** | **1.08%** | 2.70s |
>
> ***
>
> **Table: Performance comparison across RB-LARGE, ER-700-800, and SATLIB benchmarks for the Maximum Independent Set (MIS) problem.** Obj. denotes the average node number of independent sets (higher is better). Gap indicates the percentage deviation from the optimal solution (lower is better). Time is the average inference time in seconds. $S_{cand}$ is the candidate number of full DM proposals $\mathcal{P}_{DM}$.
>
> ***
>
> | METHOD | RB-LARGE&nbsp;&nbsp;&nbsp;&nbsp; Obj. $\uparrow$ | RB-LARGE&nbsp;&nbsp;&nbsp;&nbsp; Drop $\downarrow$ | RB-LARGE&nbsp;&nbsp;&nbsp;&nbsp; Time $\downarrow$ | ER-700-800&nbsp;&nbsp;&nbsp;&nbsp; Obj. $\uparrow$ | ER-700-800&nbsp;&nbsp;&nbsp;&nbsp; Drop $\downarrow$ | ER-700-800&nbsp;&nbsp;&nbsp;&nbsp; Time $\downarrow$ | SATLIB&nbsp;&nbsp;&nbsp;&nbsp; Obj. $\uparrow$ | SATLIB&nbsp;&nbsp;&nbsp;&nbsp; Drop $\downarrow$ | SATLIB&nbsp;&nbsp;&nbsp;&nbsp; Time $\downarrow$ |
> | :--- | :---: | :---: | :---: | :---: | :---: | :---: | :---: | :---: | :---: |
> | KaMIS | 43.00 | 0.00% | 56.97s | 44.97 | 0.00% | 60.75s | 425.95 | 0.00% | 24.37s |
> | Gurobi | 42.19 | 1.83% | 33.84s | 38.78 | 13.75% | 60.49s | 425.92 | 0.01% | 13.47s |
> | LWD (RL) | 36.67 | 15.31% | **0.85s** | 39.05 | 13.16% | **0.64s** | 421.80 | 1.04% | 0.66s |
> | prefix_Coexpander (DM, s=1) | 40.05 | 6.84% | 1.15s | 40.18 | 10.65% | 0.71s | 421.84 | 0.96% | **0.25s** |
> | prefix_Coexpander (s=5) | 40.06 | 6.83% | 1.22s | 40.85 | 9.16% | 0.96s | 424.17 | 0.42% | 0.91s |
> | Coexpander (s=64) | 40.58 | 5.63% | 4.48s | 40.91 | 9.03% | 4.03s | 424.78 | 0.27% | 4.27s |
> | **SpSCO ($S_{cand}$=5)** | **40.73** | **5.28%** | 2.69s | **42.38** | **5.76%** | 4.69s | **425.40** | **0.12%** | 6.31s |
>
> ***

---

> ### Author Response · Authors · 2025-11-24
> **Global Response - Practical and theoretical rationale of the single-trigger design**
>
> ## Practical and theoretical rationale of the single-trigger design
> **Practical implications:** From the practical aspect, activating the trigger multiple times could potentially yield high-quality solutions (similar to exhaustive search). However, doing so would cause the inference time to grow linearly w.r.t. the number of triggers, eliminating the efficiency advantage. We have actually experimented with multiple triggers and found that they yield marginal performance improvements, but much more inference time, as demonstrated in the following table (the multiple-trigger SpSCO is allowed to intervene up to a maximum of 20 times).
>
> ***
>
> | Setting | TSP-50 gap $\downarrow$ | TSP-50 time $\downarrow$ | TSP-100 gap $\downarrow$ | TSP-100 time $\downarrow$ | TSP-500 gap $\downarrow$ | TSP-500 time $\downarrow$ | TSP-1000 gap $\downarrow$ | TSP-1000 time $\downarrow$|
> | :--- | :---: | :---: | :---: | :---: | :---: | :---: | :---: | :---: |
> | single | 0.03% | 0.25s | 0.02% | 1.2s | 3.56% | 1.20m | 4.58% | 2.43m |
> | multiple | 0.02% | 2.38s | 0.01% | 23.28s | 1.37% | 6.43m | 4.47% | 14.22m |
>
> ***
>
> **Theoretical insights:** From the theoretical aspect, our single-trigger strategy is grounded in the theoretical optimality of minimizing total trajectory error. By treating the sequence of steps executed by the RL agent as an effective episode length $t$, established theoretical analyses in Model-Free RL [1-3] suggest that cumulative regret scales polynomially with the horizon. This results in a super-linear growth in cumulative error (e.g., $\propto t^{3/2}$) due to compounding variances and distributional shifts, causing the marginal risk of the RL policy to increase monotonically over time.
>
> In contrast, while the diffusion model's cumulative regret scales linearly with the problem dimension $t$, its marginal regret remains constant relative to the dimension $t$ [4] (All details and theorems are added in Section 4.2 and Appendix C). As visualized in Fig. 3 in the revised PDF, the total accumulated error corresponds to the area under the marginal error curves (depicted by the shaded regions). Geometrically, the intersection point $t^*$ represents the precise moment where the RL's rising marginal cost exceeds the DM's baseline. Switching policies anywhere else would inevitably increase the total shaded area, leading to suboptimal performance.
>
> Combining both the practical and theoretical considerations, SpSCO is designed to detect this specific theoretical minimum—where error propagation renders the RL policy less reliable than the DM—and execute a single, decisive handoff to ensure the optimal solution quality.
>
> Reference:
>
> [1] Jin, C., Yang, Z., Wang, Z., & Jordan, M. I. (2020). Provably efficient model-free reinforcement learning with linear function approximation. COLT.
>
> [2] Shani, L., Efroni, Y., & Mannor, S. (2020). Reinforcement learning with logarithmic regret and policy switches. ICML.
>
> [3] Ghosh, A., Zhou, X., & Shroff, N. (2022). Provably efficient model-free constrained RL with linear function approximation. NeurIPS.
>
> [4] Montanari, A. (2023). Nearly d-linear convergence bounds for diffusion models via stochastic localization. ICLR.

---

### Author Response · Authors · 2025-12-03
**Discussion Summary and Clarifications**

Dear AC,

Thank you for overseeing the review of our submission. We sincerely hope you might revisit our work, as we believe the current mixed scores stem largely from a divergence in perspective regarding the definition of 'practicality' in NCO and a misunderstanding of the theoretical grounding behind our design choices.

Before addressing specific concerns, we would like to reiterate the core contributions that define SpSCO as a distinct advancement in Neural Combinatorial Optimization:

* **A Novel Speculative Framework:** We introduce the first framework, effectively adapting speculative sampling to NCO, synergizing the speed of autoregressive drafting with the global quality of diffusion verification to transcend the inherent limitations of singular RL and DM paradigms mentioned in the abstract.
* **Theoretical Guarantee and Experimental Support:** Unlike heuristic patches, our single-trigger design guarantees a cumulative trajectory error **strictly lower than that of either standalone backbone while accelerating DM’s inference time**. This is established through a rigorous analysis of the distinct error dynamics between RL and Diffusion models. And our experimental results on large-scale settings also demonstrate the theoretical analysis.
* **Superior Speed-Quality Trade-off:** SpSCO achieves near-optimal results on small scales (0.02% gap on TSP-100) and dominates baselines on large scales, a capability now validated across diverse domains, including TSP, OP, and MIS. In the meantime, SpSCO attains faster inference speed on large-scale settings (e.g., 6x faster than T2T on TSP-1000).

While reviewers (rLv2, nQiK, sc5a) acknowledged the novelty of adapting speculative sampling to NCO and recognized the "promising direction" of addressing the speed-quality trade-off, some concerns regarding the "Single-Trigger" mechanism and the reliance on Supervised Learning (SL) persist. We believe we have addressed these in our rebuttal and extensive new experiments:

1. The Rationale of "Single-Trigger" Design: Theoretically inspired, Not Ad-Hoc

Reviewers CJH2 and sc5a initially perceived our single-trigger mechanism as an "ad-hoc" simplification compared to multi-token verification in LLMs. We clarified that this is a theoretically derived optimum specific to the NCO domain, not a heuristic shortcut.
Our additional theoretical analysis (Appendix C) proves that the RL agent’s marginal regret increases monotonically ($\propto \sqrt{t}$) while the DM’s remains constant, creating a unique intersection point $t^*$ where cumulative error is minimized. Empirically, we demonstrated that enforcing multiple triggers yields negligible gains (e.g., merely 0.01% on TSP-100) but incurs a prohibitive 10x-20x latency penalty, which defeats the purpose of speculative sampling.

In summary, we propose a mathematically grounded adaptation that organically aligns the two models based on their distinct error dynamics. (The complete analysis is in the response named *SpSCO Design Philosophy: A Theoretically Grounded Speculative Framework, Not a Heuristic Fallback*.)

2. Practicality and the Role of Supervised Learning (SL)

Reviewer CJH2 raised a concern that relying on SL backbones undermines the "data-free" practicality of RL. We respectfully argue this view overlooks the primary bottleneck of current generative solvers: Inference Latency.

We would like to highlight that our starting point is to improve DM-based neural solvers. The motivation for using RL-based approaches is to improve the test time trade-off between performance and efficiency, not the label-free training paradigm of pure RL. Diffusion models offer superior quality but are currently unusable in real-time scenarios due to slowness. SpSCO addresses this specific "practicality" gap. By accelerating inference (e.g., 6x faster than T2T on TSP-1000) without retraining, we make high-performance generative models viable for deployment.

3. Generalizability and Fair Comparison

To address concerns about scope (Reviewers rLv2, nQiK), we significantly expanded our evaluation beyond TSP.

* We added experiments on the OP and MIS. SpSCO consistently outperforms baselines even in these constrained and graph-covering tasks, proving it is a robust, model-agnostic framework.

* We addressed fairness concerns by comparing SpSCO against baselines with massive sampling. SpSCO strictly dominates these baselines in both optimality gap and inference time.

* When integrated with stronger backbones like LEHD, SpSCO outperforms recent SOTA methods like NGS in both quality (1.57% vs 2.06% gap) and speed ($\sim$4x faster), establishing a new efficiency benchmark for constructive solvers.


SpSCO offers a principled, theoretically grounded solution to the speed-quality dilemma in NCO, bridging the gap between fast RL and high-quality Diffusion models. We hope these clarifications help in your final assessment.

Thank you again for your time and support.

Warm regards,

Authors.

---

### Note · Program_Chairs · 2026-01-17
**Submission Desk Rejected by Program Chairs**

The following references in this submission do not refer to real documents and/or have major errors in bibliographic information:

 1. Ben Weinberg and J. M. K. Welling. Score-based generative models for np-hard combinatorial optimization. arXiv preprint arXiv:2106.05831, 2021.
2. Roch G. Athaide, K. S. Sesh Kumar, and S. Anil Kumar. A denoising diffusion probabilistic model for the travelling salesman problem. arXiv preprint arXiv:2302.06219, 2023.
3. Zihan Chen, Tianyu Wang, Yutong Wang, Ming-Xuan Wu, Wentao Wang, and Gaoang Wang. GPSVRP: A graph-based partition-and-search framework for vehicle routing problems with diffusion model. arXiv preprint arXiv:2309.16016, 2023b.